

# The Coastal Observing System for Northern and Arctic Seas (COSYNA)

**B. Baschek[1], F. Schroeder[1], H. Brix[1], R. Riethmüller[1], T.H. Badewien[2], G. Breitbach[1], B. Brügge[3], F. Colijn[1], R. Doerffer[1], C. Eschenbach[1], J. Friedrich[1], P. Fischer[4,], S. Garthe[5], J. Horstmann[1], H. Krasemann[1], K. Metfies[4], N. Ohle[6], W. Petersen[1], D. Pröfrock[1], R. Röttgers[1], M. Schlüter[4], J. Schulz[2], J. Schulz-Stellenfleth[1], E. Stanev[1], C. Winter[7], K. Wirtz[1], J. Wollschläger[1], O. Zielinski[2], and F. Ziemer[1]**

[1]{Institute of Coastal Research, Helmholtz-Zentrum Geesthacht, Germany}

[2]{Institute for Chemistry and Biology of the Marine Environment, University of Oldenburg, Germany}

[3]{Federal Maritime and Hydrographic Agency, Germany}

[4]{Alfred Wegener Institute, Center for Polar and Marine Research, Germany}

[5]{Research and Technology Centre (FTZ), University of Kiel, Germany}

[6]{Hamburg Port Authority, Germany}

[7]{MARUM, Center for Marine Environmental Sciences, Bremen University, Germany}

Correspondence to: Burkard Baschek (Burkard.Baschek@hzg.de)

## Abstract

The Coastal Observing System for Northern and Arctic Seas (COSYNA) was established in order to better understand the complex interdisciplinary processes of northern seas and the arctic coasts in a changing environment. Particular focus is given to the German Bight in the North Sea as a prime example for a heavily used coastal area, and Svalbard as an example of an arctic coast that is under strong pressure due to global change.



The automated observing and modelling system COSYNA is designed to monitor real time
conditions, provide short-term forecasts and data products, and to assess the impact of
anthropogenically induced change. Observations are carried out combining satellite and radar
remote sensing with various *in situ* platforms. Novel sensors, instruments, and algorithms are
developed to further improve the understanding of the interdisciplinary interactions between
physics, biogeochemistry, and the ecology of coastal seas. New modelling and data
assimilation techniques are used to integrate observations and models in a quasi-operational
system providing descriptions and forecasts of key hydrographic variables. Data and data
products are publically available free of charge and in real time. They are used by multiple
interest groups in science, agencies, politics, industry, and the public.

## 1   Introduction

A large part of humanity lives near the coasts and depends on the coastal oceans. At the same
time, global problems such as climate change, sea level rise, or ocean acidification influence
the ecosystems and communities along the coasts in particular. Shelf seas host unique
ecosystems and provide essential sources for life in the ocean and the bordering land. At the
same time, regions like the North Sea are heavily used for a multitude of human activities,
from tourism and ship traffic to the exploitation and exploration of food resources, energy and
raw materials. Shelf seas are also heavily influenced by terrestrial processes as they are
subject to a continuous influx of natural and anthropogenic material from river systems and
the atmosphere. They therefore act as important interfaces for global material cycles, for
example through the uptake, emission, and transport of carbon compounds. These regions
thus influence the Earth system and are, in turn, shaped by global change and local human
resource use. Understanding coastal systems is therefore of a high value, not only from a
scientific point of view, but also due to its societal value. Coastal research has, however, long
been hampered by the effort involved in investigating the highly complex coastal systems, the
diversity of disciplines and institutions involved, and the difficulties in obtaining long-term
and high-resolution, consistent measurements.
Current observations in the North Sea reveal substantial changes in biogeochemistry and food
webs accompanied by the occurrence of new and the disappearance of established species
(Gollasch et al., 2009; Buschbaum et al., 2012). The causes for these shifts are only partially
known. Changes in physical quantities (e.g. temperature, wind) as well as anthropogenic
influences (e.g., pollution, over-fishing, invasive species) most probably act as major drivers



(Emeis et al., 2015). In the Arctic, the thawing of permafrost has started to cause coastal erosion and an increase of greenhouse gas emissions (IPCC, 2014). These examples highlight the sensitivity and dynamic behavior of such complex systems that are still barely understood and insufficiently documented and monitored.

Recent advances in technology enable the scientific community to use their resources more efficiently by taking remotely controlled automated measurements and by developing 'intelligent' integrated systems that combine measurements and numerical modeling to create a synoptic view of coastal systems. The Coastal Observing System for Northern and Arctic Seas (COSYNA) has been established to demonstrate the feasibility of this idea for shallow, coastal areas. COSYNA focuses on the complex interdisciplinary processes of Northern Seas and the Arctic coast, to assess the impact of anthropogenic changes, and to provide a scientific infrastructure. The core of COSYNA is an extensive network of the most diverse measurement devices in the German Bight delivering near-real time data that are integrated in numerical models and are publically provided.

The principal objective of observations, instrument development, and modeling is to improve our understanding of the interdisciplinary interactions between physical, biogeochemical, and ecological processes in coastal seas, to investigate how they can be best described at present, and how they will evolve in the future. In COSYNA, data and knowledge tools are developed and provided to be of use for multiple interest groups in industry, agencies, politics, environmental protection, or the public. These data and products are publically available free of charge and can be used to support national monitoring authorities to comply, for example, with the requirements of the European Water Framework Directive and the Marine Strategy Framework Directive. The coastal observatory involves national and international contributions to international programs, such as the coastal module of the global ocean observing system (coastal GOOS), the Global Earth Observations System of Systems (GEOSS), Marine Geological and Biological Habitat Mapping (GEOHAB), and COPERNICUS Marine Environment Monitoring Service (CMEMS).

COSYNA is coordinated by the Helmholtz-Zentrum Geesthacht (HZG), Germany, and has been jointly developed, implemented, and operated with the German partner institutions Alfred Wegener Institute, Helmholtz Centre for Polar and Marine Research (AWI), MARUM, Center for Marine Environmental Sciences at Bremen University, the Institute for Chemistry and Biology of the Marine Environment at the University of Oldenburg (ICBM), the Research



and Technology Centre at the University of Kiel (FTZ), the German Federal Maritime and Hydrographic Agency (BSH), the Centre for Marine and Atmospheric Sciences at the University of Hamburg (ZMAW, now Center for Earth System Research and Sustainability, CEN), the Hamburg Port Authority (HPA), the Lower Saxony State Department for Waterway, Coastal and Nature Conservation (NLWKN), Schleswig-Holstein's Agency for Coastal Defence, National Parks, and Marine Conservation (LKN) and the German Federal Waterways Engineering and Research Institute (BAW).

This article provides an overview of COSYNA, its observational and modelling approach as well as the diverse associated scientific studies and activities. More details are provided in the contributions to this volume. This article aims at connecting them to previously published results from COSYNA. To this end, we will first describe the focus regions, objectives, and the international context of COSYNA, before giving an overview of the observations, sensor and instrument development, as well as modeling and data assimilation activities. Data, data products, and outreach activities are then described before a brief outlook over future activities is given.

## 2  Coastal focus regions

Northern and Arctic Seas are characterized by a variety of different geographical and oceanographic settings, harbour various ecosystems, and are shaped and influenced by a multitude of human uses. The focus regions of COSYNA, the German Bight of the North Sea and the Arctic coast at Svalbard, are representative for two extremes of this broad spectrum. The German Bight is one of the most intensely used coastal seas worldwide with often opposing interests of economy, nature conservation, and recreation. Arctic seas and coasts are among the areas mostly affected by and vulnerable to global warming.

### 2.1  The North Sea

The North Sea (Fig. 1) is a prime example for a shallow, heavily used coastal sea. It is a temperate, semi-closed shelf sea ranging from 51°N to 62°N. It is very shallow in the German Bight with water depths of less than 40 m. The Norwegian Trench is with 700 m depth the deepest part of the North Sea (Otto et al., 1990).

The German Bight is located at the south-eastern corner of the North Sea. Its seaward boundaries are at 6°30'E and 55°N. The main topographical features are the glacially formed



Elbe River valley that spreads out to the northwest and a chain of barrier islands along the
Dutch, German, and Danish North Sea coast. The Wadden Sea is located between these
barrier islands and the mainland. Its back-barrier intertidal flats are protected by the islands
and are separated by tidal inlets. The Wadden Sea is the largest unbroken system of intertidal
sand and mud flats in the world and is an UNESCO World Natural Heritage Site since 2009.
The North Sea is characterized by the transition from oceanic to brackish water with variable
fresh water input at the coasts. Physical drivers such as wind, sea surface temperature (SST),
or tides control the natural variability in circulation and exchange processes with the open
Atlantic and coastal fringe boundaries over a broad range of temporal and spatial scales
(Schulz et al., 1999; Sündermann et al., 1999; Emeis et al., 2015). Global and local
anthropogenic impacts overlay and interfere with these natural forcings.
Strong tidal currents and intermittent strong wind events form a regime of high kinetic and
turbulent energy with significant bed-water column exchange in the North Sea. The currents
are dominated by the M2 lunar tide that is entering the North Sea from the north and is
moving as Kelvin wave cyclonically through the North Sea (Otto et al., 1990; Howarth,
2001). Tidal currents are particularly strong in the channels connecting the Wadden Sea with
the North Sea driving an intense exchange and a net import of suspended particulate matter
and nutrients into the Wadden Sea (Burchard et al., 2008; Staneva et al., 2009; van Beusekom
et al., 2012). The tides thus cause a complex pattern of mixing conditions just off the barrier
islands and the mouths of the estuaries of the rivers Elbe, Weser, and Ems.
During each tidal cycle, typically 50 percent of the water volume of a tidal catchment area is
transported into and out of the Wadden Sea. This periodic exchange with the German Bight is
essential for the functioning of the Wadden Sea ecosystem: water entering the Wadden Sea
from the German Bight contains fine-grained sediments and particulate nutrients sustaining
the muddy component and the high productivity of the intertidal mud flats (Postma, 1984; van
Beusekom et al., 1999; van Beusekom and de Jonge, 2002).
Wind forcing is the second-most dominant factor and is particularly important during storms,
when it can generate a current response up to 25 m deep within a few hours (Howarth, 2001).
Winds in the North Sea are typically westerlies, but variations exist and southerlies and
easterlies may produce secondary circulation patterns (Otto et al., 1990).
The residual coastal current is a result of the combined effect of wind, topography, and the
density distribution (Backhaus, 1980). Its anticlockwise direction can be temporarily reversed





during predominant easterly winds (Hainbucher et al., 1987). The flushing time of the North
Sea is with 10 to 56 days (Lenhart and Pohlmann, 1997) relatively long in spite of its shallow
depth.
Strong and variable vertical and horizontal thermohaline gradients are generated by
atmospheric energy exchange and fluvial discharge. They cause the formation of a dynamic
balance of buoyancy gradients, flow-induced instabilities, and turbulence leading to features
such as fronts, filaments, and eddies (Dippner, 1993; Schrum, 1997; Sündermann et al.,

8    1999).

The North Sea is surrounded by densely populated, highly-industrialized countries and is
directly affected by multiple, often conflicting uses. In particular, the massive construction of
offshore wind farms – under way or planned – is likely to have a significant impact on marine
mammals (Koschinky et al., 2003), seabirds (Garthe and Hüppop, 2004; Busch et al., 2013),
but possibly also mixing (Lass et al., 2008; Ludewig, 2015; Carpenter et al., 2016) and
nutrient transport. The mixing region behind the barrier islands is exposed to an import of
pollutants and nutrients from land. The high biomass production caused by the latter resulted
in the identification of the entire German Bight as a problem area by the OSPAR commission
(OSPAR, 2008). Other economic exploitation activities also affect the ecosystem, such as
overfishing with bottom trawls impacting benthic invertebrate communities as well as leading
to a decrease of biomass and species richness of fish communities (Emeis et al., 2015). One of
the densest ship traffic lines worldwide crosses the German Bight and demands regular
dredging of shipping channels and harbour basins. The impact is further enhanced by near-
coast material dumping, but also sand and gravel extraction (de Groot, 1996).
The anthropogenic impact is further enhanced by the global increase of $CO_2$-concentrations
that led to a long-term increase of SST that accelerated to 0.08°C $a^{-1}$ in the last decade
(Loewe, 2009), while the average annual sea level rise reached 1.6 mm $a^{-1}$ for the last 110
years (Wahl et al., 2013), and the average pH decreased from 8.08 to 8.01 in the years 1970 to
2006 (Lorkowski et al., 2012).
**2.2   The Arctic Coast**
While Svalbard (79°N) is geographically classified as fully arctic, it is significantly
influenced by Arctic and Atlantic water masses from the Fram Strait (Hop et al., 2002).
Especially the west coast of Svalbard is alternatively exposed to warmer saline Atlantic water





masses from the West Spitsbergen Current and by colder less saline Arctic water from the
East Spitsbergen Current, or a mixture of both (Cottier et al., 2005). This bi-modal
hydrography is the basis of a complex temperate-polar balance affecting the coastal
hydrography and the associated fjord ecosystems (Svendsen et al., 2002). Due to an increased
advection rate of warmer Atlantic water masses in the fjord systems over the last decade, first
signs of an overall warming of the fjord systems have been observed with a decrease in
seasonal ice coverage (Stroeve et al., 2007) and significant changes throughout the food web
(Hegseth et al., 2013; Van de Poll et al., subm.; Willis et al., 2006, Brand and Fischer, subm.).
The 20 km long Kongsfjord is located at the west coast of Svalbard and opens to a shelf
system in westerly direction. It has no sill and shares the outlet to the Atlantic with the more
northern Krossfjord (Cottier et al., 2005). From this outlet, an underwater canyon runs
through the shelf to the continental edge, establishing a connection to the deeper waters
masses of the West Spitsbergen Current off the shelf. Complex mixing processes between the
arctic shelf water masses, the Atlantic deep water masses, and the highly seasonal fresh water
runoff from the inner part of the fjord result in strong environmental gradients from the inner
parts of the fjords to its mouth (Svendsen et al., 2002). These gradients and their short- and
long-term variability may be directly influencing the pelagic and benthic realms of the fjord
and thereby the local food web with its high spatial and temporal dynamics and complexity
(Stempniewicz et al., 2007). Due to these extremely condensed temporal and spatial patterns
of Atlantic and polar realms in a single fjord system, as well as the observed increase in mean
water temperatures, the retreat of glaciers, and decrease in sea ice coverage over the last
decades, the Kongsfjord ecosystem became an international focal point of climate change
research.
The Kongsfjord (Fig. 2) is one of the best studied fjord systems of the west coast of
Spitsbergen. The first research station was built by the Norsk Polar Institute in NyÅlesund
(Fig. 3) at 78°55'N, 11°56'E in 1970. Since then, more than 15 nations operate their own
research stations in this northernmost year-round inhabited research-village of the world
including the German-French research station AWIPEV (www.awipev.eu).
Even in Kongsfjord with its ideal and year-round available research infrastructure, most field
research has been done so far in summer during the polar day (Fischer et al., this issue) and
only very little is known about the polar winter with its prevailing darkness during a period of
several months. The winter months are, however, essential for life cycles, the reproduction of


many species (Fischer et al., this issue), and hence for the entire ecosystem (Hop et al., 2012).
It is COSYNA's aim to help close this observational gap providing year-round observations
in the Kongsfjord.
COSYNA activities also comprise remote sensing techniques, that have been proved and
tested in the North Sea, to coastal waters in the Lena Delta, Siberia for the quantification of
suspended matter and chlorophyll as well as *in situ* measurements of inherent optical
properties (Örek et al., 2013). The Lena Delta covers 32.000 km$^2$ and discharges freshwater
from a catchment area of 2.400.000 km$^2$ into the Arctic Ocean.
**3    Objectives and Benefits**
Complex, highly interdisciplinary natural processes characterize the North Sea across several
time and length scales. It is COSYNA's goal to help disentangle natural processes and
anthropogenic impact in this region by combining consistent long-term time series at
representative locations with process-oriented high-resolution observations. Numerical
models are used to integrate observations ranging from the turbulent to basin wide spatial
scales while bridging time periods from minutes to decades. It has therefore been COSYNA's
approach to build an integrated observing system that is geared towards high flexibility and
can be used on a variety of scales and problems that are of scientific or societal interest.
Routine observations of key variables and data assimilation techniques are employed to
improve model performance for hindcasts, nowcasts, and short-term forecasts. The
implementation of such a system achieves several objectives: it bridges spatial and temporal
scales, while it establishes a backdrop against which key processes, such as exchange
processes between North Sea and Wadden Sea, the impact of extreme events, biological
productivity variations, and the influences of e.g. offshore wind farm construction can be
investigated. The extensive development of offshore wind farms, for instance, requires sound
environmental statistics and improved forecasts for planning and operation, while their
influence on hydrodynamics, let alone biogeochemistry or biology, of the North Sea is still
poorly understood.
The benefits of the COSYNA system are expected to be manifold. It contributes to
technology development of key sensors and infrastructure, data interpretation algorithms such
as for satellites and radar, as well as to modelling and data assimilation techniques suitable for
operational use and monitoring. These developments and the creation of products of interest





for various user groups contribute to the sciences while also benefitting society, for example,
through supplying coastal and sea floor observations for the North Sea in support of the
European framework strategies and directives towards the goal of achieving a "good
environmental status" of the marine environment.
As for the dissemination of data and products, COSYNA's objective is to make them
available free of charge to the broadest possible audience in near-real-time, while ensuring
high quality standards and rigorous monitoring of data quality. Additional quality controls
taking long-term perspectives into account are to be performed on an on-going basis
ultimately resulting in data publications.

## 4    International Context

With the initiation of the permanent Global Ocean Observing System GOOS
(Intergovernmental Oceanographic Commission, 1993) and stepwise implementation of its
many separate observing systems, new concepts regarding the world-wide systematic and
sustained observation of the oceans have been put in place. Considering the role of coastal
areas for ecological communities and their exposure to massive human utilization, a GOOS
coastal module was proposed to provide a basis for extended predictability of the coastal
environment in both model and observations (Intergovernmental Oceanographic Commission,
1997). Awareness of the multitude of societal benefits (ABARE, 2006;
https://ioos.noaa.gov/about/societal-benefits/) stimulated considerable investment into the
worldwide implementation of integrated coastal ocean observatories (ICOOS). The United
States of America, for instance, coordinate their ICOOS within Regional Associations of the
U.S. IOOS (U.S. IOOS Office, 2010) as their GOOS Regional Alliance (GRA) contribution.
The Australian Integrated Marine Observing System IMOS (Moltmann et al., 2010) is another
prominent example for a GRA that comprises numerous observational and modelling
subsystems to generate coherent operational products from the coastline to the deep ocean
surrounding Australia. For IMOS, a detailed study (ABARE, 2006) estimated a total annual
benefit of AU$ 615 million and a benefit to cost ratio of more than 22.
In Europe, EuroGOOS (http://eurogoos.eu/) is the pan-European GRA that co-ordinates six
regional operational systems (ROOSes), such as the North West Shelf Operational
Oceanographic System (NOOS, http://eurogoos.eu/roos/north-west-european-shelf-
operational-oceanographic-system-noos/). In addition to providing operational oceanographic
services and carrying out marine research, EuroGOOS puts considerable effort into unlocking



fragmented and hidden marine data and making them openly available. Its data plays a key
role in the development of the European Marine Observation and Data Network (EMODnet)
data portals (http://www.emodnet.eu/). EMODnet is designed to cover all European coastal
waters. The European ROOSes feed data into EMODnet either directly, or for physical data,
exploiting the infrastructures and services from SeaDataNet (Schaap and Lowry, 2010;
http://www.seadatanet.org/) and the Copernicus Marine Environment Monitoring Service
(CMEMS, http://marine.copernicus.eu).
COSYNA contributes through the Helmholtz-Zentrum Geesthacht (HZG), as EuroGOOS
member, to the definition and implementation of operational services for near coast, shallow
ocean waters. Based on the FerryBox project funded by the EU in 2002-2005, HZG is co-
chairing the FerryBox EuroGOOS Task Team (http://www.ferrybox.org). Via NOOS, the
FerryBox data are fed into the EMODnet portals, while COSYNA's High-Frequency radar
data are delivered directly to the EMODnet Physics data portal and the glider data to the
CMEMS data server.

## 5   Observations

The COSYNA observation network was designed to cover spatial scales ranging from a tidal
catchment area in the Wadden Sea to the southern North Sea (Fig. 1). An additional observing
station was installed in the Arctic at the west coast of Svalbard. Nearly all platforms are
equipped with instruments to deliver a set of COSYNA standard observables comprising key
meteorological, oceanographic, and biogeochemical bulk parameters (Table 1). Tables 2 and
3 provide a comprehensive overview of the COSYNA platforms.
Starting from the Wadden Sea coast line, four stationary systems were installed on poles
placed in tidal basins of the East Frisian (3) and North Frisian (1) Wadden Sea. The tidal
dynamics are resolved to allow the estimation of energy and matter budgets over the sampled
catchment areas. An additional pole and a stationary FerryBox monitor the exchange between
the German Bight and the Elbe river as its main tributary.
To estimate transports across the northern cross-section of the German Bight, a FerryBox was
installed on the wind-turbine research platform FINO3 (Forschungsplattformen in Nord- und
Ostsee). Upstream of it, along the mean transport pathway in the German Bight, the FINO1
platform is located, where the German Federal Maritime and Hydrographic Agency (BSH)
operates one station of its Marine Environmental Monitoring Network in the North Sea and


Baltic Sea (MARNET). In general, MARNET complements the fixed COSYNA platforms
(Table 2) towards the offshore regions of the German exclusive economic zone (EEZ).
To extend the COSYNA Network to the North Sea-scale, FerryBox systems are operated on
several ships of opportunity, with regular routes between the German mainland and the island
of Helgoland and between Germany, England, and Norway.
To provide a good spatial coverage of some oceanographic and biogeochemical parameters,
remote sensing with high frequency (HF) radar and satellites is used. A HF radar array is
installed at the East (1) - and North (2) Frisian coast. Their nearly rectangular viewing angles
allow the determination of horizontal surface current vectors over most of the German Bight.
The surface concentrations of total suspended matter, chlorophyll-a, and yellow substances
"Gelbstoff" were obtained from 2003 to 2012 with MERIS (Medium Resolution Imaging
Spectrometer) onboard ENVISAT, followed by MODIS (Moderate Resolution Imaging
Spectroradiometer).
To go beyond the limitations in power and data transmission rates that most COSYNA
platforms face, two COSYNA Underwater-Node Systems were developed and installed. They
are pilots towards long-term observations of parameters beyond the COSYNA standard
observables, such as optical systems for non-invasive determination of plankton or fish
populations and their behavior. The underwater node off the island of Helgoland is the first
installation in a shallow water environment worldwide subject to strong wave forces. At
Svalbard, the underwater node allows year-round observations under the sea ice in harsh
environmental conditions. To explore physical and biogeochemical processes at the sediment-
water interface over longer periods of time in high detail, three lander systems were
developed, that can be connected to the Underwater-Node Systems for longer operations.
Observations of the vertical distribution of variables over most of the water column were
achieved with two alternating gliders operating for several weeks north-west off the island of
Helgoland. Ship cruises with an undulating towed fish were carried out two to four times per
year along a repeated grid covering the German Bight with the MARNET stations at its
crossing points. For details on the moving platforms used in COSYNA see Table 3.
All data are transferred in near-real time to the COSYNA data server and are publically
available in the COSYNA data portal (CODM). Quality control processes are applied and
data are flagged accordingly.



## 5.1 Fixed-Point Measurements

Fixed stations are the central element of COSYNA and serve as platforms to record point-like time series of meteorological and marine parameters. They provide high frequency observations to resolve variability well below tidal periods in order to estimate statistically significant tidal fluxes as well as long-term records over several years at the same location. Measuring poles were implemented at three tidal inlets, the inner Hörnum tidal basin, the Jade Bay, and the Otzumer Balje close to the island of Spiekeroog, to capture the hydrodynamics and suspended particulate matter concentrations (SPMC) typical of the East Frisian and North Frisian Wadden Sea. An additional pole was placed in the outer Elbe estuary (Fig. 1).

While the inner Hoernum tidal basin represents the zero usage zone of the National Park of the North Frisian Wadden Sea, the Jade Bay was exposed to intense activity of building a new deep water port. The Otzumer Balje discharges a catchment area that is typical for the East Frisian Wadden Sea and was intensely investigated during the ecosystem research project ELAWAT (Dittmann, 1999). Long-term year-round observations in the tidal inlet between the East Frisian islands of Langeoog and Spiekeroog were performed with the measuring pole Spiekeroog that was setup in 2002 as part of the research programme BioGeoChemistry. The Elbe pole was operated to contribute to the sediment management plan of the Elbe Estuary and to complement the data of the stationary Cuxhaven Ferry-Box on the southern side of the Elbe mouth. The FerryBox on FINO3 captures offshore conditions in the German Bight. All these stations are described in the following in more detail (Table 2).

### 5.1.1 Poles at Hörnum Deep, Jade Bay, and Elbe Estuary

The pole at the inner Hörnum tidal basin, the Jade Bay, and in the Elbe Estuary were mounted from March to November to prevent ice damage in the winter months. They consisted of a 15 m long steel tube, 5 m of which were jetted into the sea bed. A platform accessible via a ladder was mounted on top of the 40 cm-diameter tube, resulting in an overall length of 18 m (Fig. 4). The platform carried meteorological sensors and radiometer, solar panels for energy supply, an automated yet remotely controllable water sampler, and logger boxes for temporary data storage and wireless communication. A manual winch was used to retrieve the underwater instrument unit for maintenance. The underwater unit was mounted so that the lower end of its sensor package was positioned 1 m above the sea floor. It was equipped with





sensors for all COSYNA standard observables of physical oceanography and biogeochemistry
(Onken et al., 2007; Kappenberg et al., this issue; Table 1).
In order to reduce sensor fouling, e.g. by seaweed, mussels, barnacles and other organic
material, the underwater unit was cleaned at least twice a month. Possible sensor drift and
cleansing effects were monitored by direct comparison with a well-calibrated reference
system before, during, and after maintenance. Water samples were taken during maintenance
to relate optical signals to SPMC.
To observe heat fluxes between the tidal flats and the water body, a vertical temperature
sediment profiler was developed in the intertidal sediments close to the pole (Onken et al.,
2010) for more than a year. At a distance of 5 nautical miles, an additional mooring with an
upward looking ADCP (Acoustic Doppler Current Profiler) and a Datawell wave rider buoy
was deployed.
In order to compute along-channel fluxes in the Hoernum Deep, occasional ship surveys were
carried out over full tidal cycles relating across- and along-channel transects to the pole data.
They were complemented by additional water samples with accompanying turbidity
measurements. Fig. 5 displays an example for a Hoernum pole time series of water level,
significant wave height, wind and current speed, water temperature, salinity, and SPMC. This
period comprises a significant wind event with peak velocities up to 20 ms$^{-1}$ resulting in a sea
level rise of more than 1.5 m and significant wave heights up to 1.7 m. Water temperature and
salinity after the storm exhibit the characteristic tidal (mainly M2) variability. Current
velocities are predominantly M4, with a clear ebb-flood asymmetry. SPMC shows a complex
variability reflecting the M4 tidal current dependencies as well as horizontal along-channel
gradients. Interestingly, the onset of the rise in SPMC and its peak value lag behind the
significant wave height by nearly one tidal period indicating that the source of the additionally
suspended material is located remote from the pole.
The combination of long-term observations of near-point time series with cross-sectional ship
surveys indicates that the steady import of particulate matter is closely connected to the
specific thermodynamic processes of the amphibic Wadden Sea area (Burchard et al., 2008;
Onken et al., 2007; Onken and Riethmüller, 2010; Flöser et al., 2011). The analysis of the
Elbe pole data from the years 2012 and 2013 is described in Kappenberg et al. (this issue).





### 5.1.2 Pole Spiekeroog
Time series of oceanographic, meteorological, and biogeochemical data are continuously
recorded since 2002 at a measuring pole (Fig. 1, Fig. 4) of the Institute for Chemistry and
Biology of the Marine Environment in a tidal channel close to the island of Spiekeroog
(Reuter, 2009; Badewien et al., 2009). The time-series station Spiekeroog (position
53°45'0.10''N, 007°40'16.3''E, mean sea level 13 m) consists of a 35.5 m long pole, with a
diameter of 1.6 m that is driven 10 m into the sediment. A platform is mounted on top of the
pole, about 7 m above sea level. It consists of two laboratory containers hosting a second
platform on top that is equipped with solar panels, a wind turbine and meteorological sensor
systems. Oceanographic sensors are installed in special tubes within the pole, that are oriented
in the main direction of the tidal flow. An Acoustic Doppler Current Profiler is mounted 1 m
above the sea floor on a horizontal arm of 12 m length. The time-series station Spiekeroog is
capable of withstanding storm events and ice conditions. It is part of COSYNA since 2012.
The acquired data sets are fundamental for the improvement and validation of model results
(Burchard and Badewien, 2015; Grashorn et al. 2015; Lettman et al., 2009; Staneva et al.,
2009; Burchard et al., 2008) as well as to answer various research questions (Rullkötter, 2009;
Badewien et al., 2009; Hodapp et al., 2015; Meier et al., 2015; Holinde et al., 2015) and to
improve fouling-prone sensing methods and quality assurance (Garaba et al., 2014; Schulz et
al., 2015; Oehmcke et al., 2015).
### 5.1.3 Stationary FerryBoxes
As part of the COSYNA network, a stationary FerryBox was installed inside the pole of
research platform FINO3. Water is pumped from approx. 5 m and 16 m below mean sea level
height for the continuous analysis near-surface and sea floor waters. The FerryBox is
equipped with sensors for standard oceanographic parameters (Table 1). Temporarily, nutrient
analysers and a $pCO_2$ sensor were added.
Despite harsh operating conditions, the FerryBox is operational since July 2011, with short
interruptions during storm periods that were caused by sea spray and condensation that
occurred notwithstanding the use of a heated steel cabinet for the protection of its electronics.
Due to its remote position in the North Sea, personnel and spare parts had to be transported by
helicopter to the platform for maintenance. Weather conditions therefore constrained the



accessibility of the platform and sensors requiring regular maintenance could only be used
temporarily. The software was operated remotely.
Since August 2010, a stationary FerryBox is also installed in a container directly at the
waterfront of Cuxhaven Harbour. It samples the tidally influenced, highly turbid lower Elbe
river, the main freshwater discharge into the COSYNA observation area. The FerryBox was
complemented by the Elbe estuary measurement pole located 18 km upstream on the northern
side of the river (Section 5.1.1) to contribute to a better understanding of the SPM dynamics
and transport through the Elbe estuarine turbidity zone into the German Bight.
The water intake is located at a mean depth of 4 m. The standard oceanographic sensors are
described in Section 5.4. The FerryBox is also equipped with a nitrate, phosphate, and silicate
analyser as well as a fluorescence-based instrument for phytoplankton group determination. A
meteorological station mounted on the top of the container provides wind speed and global
radiation values.
Due to its easy and constant accessibility, the FerryBox Cuxhaven is an ideal platform for the
testing of the long-term performance of new sensors under environmental conditions.
As example, a time-series of several parameters is shown for 2012 and 2013 (Fig. 6). A strong
discharge period in summer of 2013 led to a substantial decrease of salinity with nearly fresh
water conditions at low water for a two week period (Voynova et al., this issue).

## 5.2  Ocean Gliders

Ocean gliders (Fig. 7) are autonomous underwater vehicles, propelled by a buoyancy engine.
In the last decade they have become an established oceanographic platform in the open ocean
autonomously collecting data with a high temporal resolution along (re)programmable
transects. Due to their operational flexibility and a long endurance on the order of months,
gliders sample the oceans at low cost in a way no other platforms currently do (Testor et al.,
25 2010).

The use of ocean gliders in shallow coastal waters is, however, challenging. COSYNA and a
few other observatories have pioneered this particular use. Due to bathymetric constraints,
currents can reach magnitudes in excess of the nominal glider speed, making it difficult to
follow a prescribed transect. Intense commercial and recreational shipping traffic significantly
increases the likelihood of a glider-ship collision (Merckelbach, 2013). This will almost





certainly result in the loss of the glider and possibly in a hull rupture, if a fast light-weight
craft is involved (Drücker et al., 2015). Therefore, COSYNA collaborates closely with the
authority responsible for safety regulations in the German sector of the North Sea (Wasser-
und Schifffahrtsamt) to develop prediction methodologies to mitigate the risk at sea involving
gliders (Merckelbach, this issue).
COSYNA maintains three Slocum Littoral Electric gliders (Jones et al., 2005). These gliders
have been used in the German sector of the North Sea in different operational modes. Gliders
are particularly well suited for surveying repeated transects over long periods of time
(months). Their long endurance makes it viable to run two gliders in an alternating service.
While one glider is operational, the second one is refurbished. The gliders have also been
deployed for shorter, targeted experiments. The use of multiple gliders provides additional
spatial information. In order to fly gliders in formation, operational techniques have been
developed so that they act as a single entity facilitating the interpretation of the spatial
variability. The measurements taken with COSYNA gliders is available on CODM. With the
help of a Java applet, glider data can be visualized in 3 dimensions (Breitbach et al., 2016).
The evolution of stratification during 2012 and part of 2014 is shown in Fig. 8 to illustrate
glider measurements. The data was collected by two gliders in alternating service in 2012, and
within a single experiment in 2014. From May to August, the potential energy and
stratification of the water column increases due to solar heat flux. During that time, the water
column is partially mixed by wind and waves at several instances. After September, mixing
dominates and the heat fluxes are too low to create a stable stratification. Data from 2014
shows interannual variability with a strong stratification in August and a subsequent complete
mixing of the water column caused by a storm. After this event, the stratification was not
restored.
**5.3   High-Frequency Radar System**
In order to detect surface currents, a High Frequency (HF) radar network was established in
the German Bight of the North Sea. It consists of three "Wellen Radar" (WERA) systems
(Gurgel et al., 1999) located on the isles of Sylt and Wangerooge and in Büsum (Fig. 9).
The radar signal propagates along the ocean surface beyond the horizon and is backscattered
by surface waves on the order of 10 m (half the electromagnetic wave length). The WERA
systems typically cover a range distance of 100 km with a resolution of 1.5 km. All systems





transmit via a rectangular array of four antennas with a total power of 32 W. The Systems on
Sylt and in Büsum operate at 10.8 MHz with a linear receiver array consisting of 12 antennas,
while the radar on Wangerooge operates at 12.1 MHz with a 16-antenna array.
The acquired data are subject to quality control and are publically available within 30 min of
acquisition. In an additional processing step, the radial components of each radar site are
assimilated into a numerical simulation model (Stanev et al., 2014) that is also used for short-
term forecasts.
Since 2013, the HF radar network is also used for ship detection, tracking, and fusion.
Although the HF radar network was setup for the retrieval of oceanographic parameters,
leading to a limited resolution and detection performance, ship detection can be performed at
each HF radar station every 33 s (Dzvonkovskaya et al., 2008). Tracking and fusion is
performed as a post processing task utilizing state-of-the-art algorithms (Bruno et al., 2013;
Maresca et al., 2014; Vivone et al., 2015).
**5.4   FerryBox**
In order to obtain oceanographic variables in a cost-effective way on a routinely basis,
FerryBox-systems have been developed within COSYNA and were installed on several ships-
of-opportunity such as ferries or cargo ships, research vessels, or as stationary units (Fig. 10).
They deliver key physical state variables of the North Sea and the Arctic coast off Svalbard
and fill gaps concerning robust biogeochemical observations of the oceans. In particular,
observations of the coastal carbon cycle with high temporal and spatial resolution along the
ship tracks help to understand impacts of climate change or eutrophication on productivity, as
well as the influence of single events such as storms or floods on the system. The measured
variables include temperature, salinity, chlorophyll-a, dissolved oxygen (DO), the partial
pressure of $CO_2$ ($pCO_2$), pH, alkalinity, nutrients, turbidity, or algal groups. The data are used
for model validation (Petersen et al., 2011; Haller et al., 2015) and assimilation studies
(Stanev et al., 2011; Grayek et al., 2011; Fig. 11).
The FerryBox is a modular system that can be easily extended with additional sensors.
Compared to other platforms, such as buoys, the FerryBox-systems have fewer limitations
due to space, power consumption, or harsh environmental conditions allowing the operation
of experimental and less robust sensors (Petersen, 2014).





All data are stored in the FerryBox-system and are transferred to the COSYNA server when
the vessel has a stable internet connection. COSYNA's FerryBoxes are part of an
international network within EuroGOOS (http://www.ferrybox.org).
**5.5   Underwater-Node System**
While cabled underwater observatory technology has been developed for deep sea research
applications over the last decades, cabled underwater observatories for shallow water were
only recently initiated due to the predicted dramatic effects of climate change especially in the
world's coastal regions. They are needed as core research infrastructures when either a
continuous high-frequency or real-time monitoring of hydrographical or biological data is
required or when scientific instrumentation requires more power than batteries can provide.
Cabled underwater observatories enable new research approaches in marine science by
providing long-term time series. Similar to atmospheric or terrestrial research, they are
suitable to form the backbone of international coastal and climate change research.
The harsh environments of shallow waters with extreme wave impact, storms, sea ice, strong
currents, as well as biofouling and the direct impact of fishing vessels require the
development of very robust cabled systems. COSYNA has started with this development in
2010, with the goal to observe multidisciplinary processes in the harsh environmental
conditions in the North Sea and in the arctic areas – in particular during storms and in winter
when access with vessels is difficult or impossible.
The COSYNA Underwater-Node System comprises a land based power unit and server
providing 1000 VDC, a GBit-network connection, and virtual computer technology for up to
20 different users. This land-based control system is connected to the underwater node unit
via a fibre-optic and power hybrid cable that can be up to 10 km long (Fig. 12).
The underwater unit is built as basic lander system. Up to 10 underwater plugs provide power
und network connection. The underwater unit can be outfitted with an uninterrupted low-
power battery supply for 6-8 hours operating time to enable temporary disconnection from the
high voltage electricity. From this central underwater node unit (Fig. 12-3), sensors or sensor
units with a power consumption of up to 200 W (Fig. 12-4) can be connected via an up to 70
m long cable. Communication and data transfer with the attached sensors or sensor units are
realized via TCP/IP. Completely separated ports allow scientists to directly communicate with
the instruments independent of other users. From the primary node system, an uplink power



and network connection allows the serial connection of a secondary and tertiary underwater
node unit (Fig. 12-5) to reach a maximal range of 30 km from the land based support unit.
Since 2012, COSYNA operates two Underwater-Node Systems. One node system with 10
separated ports is located off the island of Helgoland at 59° 11'N / 8° 52,79E in 10 m water
depth close to the long-term time series station "Helgoland Roads" and the AWI underwater
experimental area MarGate (Wehkamp and Fischer, 2012; 2013a; 2013b). It is operated as
permanent monitoring facility for the main hydrographical parameters in the southern North
Sea (temperature, conductivity, $O_2$, pH, turbidity, currents), as docking and support system for
complex sensor systems with high power and data transfer demands, such as stereo-optical
cameras (Wehkamp and Fischer, 2014), and as test facility for the development and operation
of the Underwater-Node Systems in the shallow environment of the North Sea. Since 2012,
the Helgoland node system endured two severe storms with wind speeds of up to 12 Bft. (190
km $h^{-1}$) providing evidence that the operation of cabled observatories is possible under
extreme conditions.
Because the North Sea experiences strong winds of more than 6 Bft. for more than 150 days
per year (Fig. 13), the cabled observatory provides an invaluable extension of ship-based
research. It may therefore help fill a significant gap in our understanding of ecosystem
behaviour in coastal environments beyond 6-8 Bft. when ship-based research is very limited
or impossible due to safety constraints.
The second continuously operated COSYNA underwater observatory is deployed since 2012
off Svalbard at 78° 92'N, 11° 9'E. It is located at the west coast of Spitsbergen close to the
international research village of NyÅlesund. It comprises a FerryBox system and a COSYNA
Underwater Node System at the "Old Pier" (Fig. 3) close to the research village of
NyÅlesund. It provides a continuous year-round monitoring system as well as an access point
for international project partners. Since 2015, the COSYNA underwater observatory is part of
the EU project Jerico-Next, the long-term research strategy of the NyÅlesund research
council, and the Kongsfjord Flagship Program.
Also the Svalbard observatory is operated as permanent monitoring facility for the main
hydrographical parameters in the fjord system (temperature, conductivity, $O_2$, pH, turbidity,
currents) and as docking and support system for complex sensor systems. It is fully remotely
controlled and all sensors and sensor units can be accessed via the internet from Germany.
The Svalbard observatory is equipped with 4 access points and is specifically designed for





national and international cooperations in the Kongsfjorden ecosystem. A main feature of the
Svalbard observatory is a vertical profiling sensor unit, which allows to remotely position
attached sensors at a specific depth on a daily or even hourly basis. Thus, the entire water
column can be sampled year-round, even under sea ice.
With the remotely controlled sensor setup of the COSYNA Underwater-Node System, it was
for the first time possible to gain data with a temporal resolution of up to 1 Hz with both,
CTD and ADCP sensors, but also with highly complex sensors like a stereo-optical camera
system that is able to measure abundance, species composition and length frequency
distributions of macroscopic organisms (Wehkamp and Fischer, 2014). No data set of this
kind has previously been available from any Arctic ecosystem worldwide, thus providing
unique insights into the polar dynamic (Fig. 14).
**5.6  Landers**
Under the COSYNA framework, different autonomous sea floor observatories (landers) have
been developed and are applied in various past and ongoing research programmes. These
landers bridge the observational gap between long term monitoring stations, remote sensing
applications, and ship-based field campaigns. They are mobile, and can be used to spatially
interpolate between monitoring stations and provide data with very high temporal resolution
(Kwoll et al., 2013; Kwoll et al., 2014; Oehler et al., 2015; Ahmerkap et al., subm.). Lander
operations aim at measuring various processes close to the sea floor or in the sediment and are
designed to have minimal impact on the environment and quantities that are measured. The
landers can be either operated autonomously for days or weeks at a time, or may be connected
to the COSYNA Underwater-Node System that is providing power and data connection for
the landers.
The landers developed and used in COSYNA are i) the lander SedObs (Sediment Dynamics
Observatory) measuring seafloor dynamics, ii) the lander NuSObs (Nutrient and Suspension
Observatory), and iii) the Lander FLUXSO (Fluxes on Sand Observatory).
5.6.1  Lander SedObs
The Sediment Dynamics Observatory (Lander SedObs) is used to investigate seafloor
dynamics and to improve the fundamental knowledge of multi-phase flows and the interaction
of physical and biological processes. The sea floor and lower water column are characterized



by morphodynamic processes acting on a large range of spatial and temporal scales.
Observations with SedObs focus on short-term dynamics due to tides or storm events.
Particular focus is given to the interaction of water motion by currents and waves as well as
the transport of sediments and other substances with the sea bed evolution under the influence
of (micro)biological stabilizing and destabilizing organisms (Ahmerkamp et al., 2015).
SedObs consists of a $2\times2$ m steel frame with a platform providing space for battery power
supply and the installation of sensors (Fig. 15). The platform rests on four adjustable and
inclined legs. Foot plates provide stable stand, prohibit subsidence, and reduce scouring
around the legs. Sensors can be attached to the legs for measurements close to the sea bed.
The lander is deployed with a launching frame from a research vessel orienting it in the
direction of main currents. After release of the lander, the frame is recovered in order to
minimize flow disturbances. For recovery, a floating buoy with recovery line is released
acoustically. Typical deployment times exceed 25 h to account for tidal variations.
Flow velocities and turbulence above and below the lander are measured with two Acoustic
Doppler Current Profilers. The upward-looking ADCP also captures the directional surface
wave spectrum. Two Acoustic Doppler Velocimeters record velocity at two levels with high
frequency. Turbulence characteristics are computed from high frequent velocity fluctuations.
The small-scale bathymetry below the lander is measured with a 3D-Acoustic Ripple Profiler
(Bell and Thorne, 1997). The sensor is installed about 1.8 m above the seafloor covering a
circular area of 6.2 m diameter. Sediment transport characteristics are measured with Sequoia
Lisst 100X instruments providing in-situ particle size distributions of suspended sediments.
Characteristics of suspended matter concentration are provided by optical backscatter sensors
and the backscattered signal strengths of the hydroacoustic instruments. Additional
parameters comprise the COSYNA standard observables. Observations are complemented by
investigations of benthic species as well as sedimentological and granulometric analysis
(Laser diffraction) of the sediments sampled with grab samplers, box corers, and multi-corer
equipment.
SedObs supports several applied and fundamental research projects, such as KÜNO NOAH
(North Sea Observation and Assessment of Habitats). Until 2015, eleven ship surveys were
carried out, field data were collected, and analysed at different reference sites in the German
Bight with sedimentological and morphological characteristics that are representative for
large areas of the German EEZ in the North Sea. A combination with other COSYNA sea



floor observatories has produced consistent and extensive data sets on various physical and (micro)biological properties of the domains (Krämer and Winter, this issue). Data are published at http://www.noah-project.de.

During some parts of the tidal cycle a periodic stratification of the water column has been observed in shallow areas of the German Bight forming distinct layers that move independently with a decoupled tidal ellipticity (Krämer and Winter, this volume; Kwoll et al., 2013; Kwoll et al., 2014; Ahmerkap et al., submitted). The difference in sea bed dynamics between fair weather conditions and storms is also investigated in the research area "Seafloor Dynamics" of the Deutsche Forschungsgemeinschaft (DFG, German Research Foundation) Research Center / Cluster of Excellence „The Ocean in the Earth System".

### 5.6.2 Lander NuSObs

The benthic lander system NuSObs (Nutrient and Suspension Observatory) was designed to quantify the exchange of nutrients and oxygen across the sediment-water-interface and to sample surface sediments *in situ* (Oehler et al, 2015a; Oehler at al., 2015b). It was the aim to study the remineralization of organic matter, the reflux of nutrients into the bottom water, the dissolution of biogenic silica (e.g. Diatoms) and, transport processes across the sediment-water transition zone, such as biologically mediated transport (e.g. bioirrigation) or wave induced pore water advection. Target area was the North Sea. Three time series sites were selected and revisited three to four times a year in order to identify seasonal variations.

NuSObs (Fig. 15) was equipped with two ″Mississippi″ type chambers (Witte and Pfannkuche, 2000). After the deployment of the lander, both chambers were moved slowly into the sediment by a motor each enclosing a sediment area of 400 cm² for typically 12–24 h. Each chamber was equipped with a syringe sampler (seven 50 ml glass syringes) to obtain water samples from the incubation chamber for subsequent chemical analysis. In addition, an oxygen Optode and pH sensor were mounted in each chamber. The syringe sampler was pre-programmed to obtain water samples from the chamber every 2–3 h yielding time series data of oxygen, nitrate, or silicic acid concentrations within the chambers.

### 5.6.3 Lander FLUXSO

The benthic lander system FLUXSO (Fluxes on Sands Observatory) was developed for studying *in situ* solute fluxes of nutrients, DIC, and oxygen in permeable consolidated





sediments. It is the goal to assess the importance of the seafloor as sink or source of nutrients
and benthic-pelagic coupling and to study advection-related processes in permeable shelf
sediments.
The lander consists of a tripod base frame that is recovered from the seafloor using two pop-
up buoys (Fig. 16). Power supply is provided by a deep-sea battery. The lander contains two
wiggling chambers that are both equipped with oxygen and $CO_2$ optodes, a pH sensor, and a
conductivity sensor. A stirrer disk with variable speed and direction allows the simulation of
advective or diffusive flow regimes in each chamber by creating rotationally symmetric
pressure gradients between the center and the circumference of the enclosed sediment surface.
The shape and magnitude of the pressure gradients closely resemble natural conditions. Two
syringe samplers are used for tracer injection and sampling from the chambers. Outside water
parameters are measured with a CTD with fluorescence and turbidity sensors, a PAR sensor,
an oxygen optode and pH sensor, as well as a Doppler current sensor.
The FLUXSO lander can be deployed at the seafloor, where it autonomously measures solute
fluxes between sediment and sea water using isolated sampling chambers. An innovative
wiggling mechanism is used, permitting gentle and deep penetration of the chambers into
consolidated sediments with minimum disturbance (Janssen et al., 2005).

## 5.7   Satellite Oceanography

Satellite remote sensing is unique in providing a synoptic view over larger areas of the sea
surface (Robinson, 2004). Standard algorithms are used widely to determine the optically
dominant water constituents and the chlorophyll concentration in clear oceanic waters (Carder
et al., 1991; Lee et al., 1998; Gohin et al., 2002). These simple band-ratio algorithms,
however, often fail in optically complex coastal waters. To gain concentrations of one coastal
water constituent, other optically active substance categories have to be considered in the
development of algorithms for the inversion of satellite spectral data. The correction of the
atmospheric influence is more sensitive and complex as it accounts for 90% to 98% of the
radiance seen at the satellite. The algorithms for coastal waters developed by HZG and used
in COSYNA are included in the ESA (European Space Agency) operational processing
scheme for the sensors MERIS (MEdium Resolution Imaging Spectrometer) on ENVISAT
(Doerffer and Schiller, 2007) and OLCI (Ocean and Land Colour Instrument) on Sentinel-3



providing chlorophyll-*a* and total suspended matter (TSM) concentrations and the absorption
by chromophoric dissolved organic matter (CDOM, "Gelbstoff").
MERIS provided COSYNA data (Fig. 17) for the North Sea until 2012 when ENVISAT
failed. With the adaption of the coastal algorithm to MODIS (on AQUA) and OLCI the
continuation of data coverage was ensured. For the validation and further improvement of
these algorithms, a series of COSYNA research cruises (Section 5.9) was conducted in 2009 -
2014 to collect optical and biogeochemical ground-truthing data.

## 8   5.8   Seabird Tracking

Seabirds are top predators depending on marine resources. Their foraging behavior may
therefore indicate changes in their food resources which are often associated with variability
in the marine environment (Furness and Camphuysen, 1997). In COSYNA, the Northern
Gannet (*Morus bassanus*) has been selected as the target seabird species due to their size and
large foraging range (Fig. 18; Garthe et al., this issue). Northern Gannets are widely
distributed in the North Atlantic and breed in large colonies. Individual Northern Gannets
were equipped with modern, lightweight GPS data loggers to track their flight patterns and
foraging behavior. In particular, information is collected on position, flight speed, altitude,
and partly also on dive depth and water temperature. A strong feature of most modern data
loggers is that they are powered by solar cells thus enabling long-term tracking for several
weeks, months, or even years. Furthermore, an increasing number of devices provide data
transfer via UHF, satellite, and mobile phone networks (Wilson and Vandenabeele, 2012;
Kays et al., 2015). A combination of the data collected by seabirds with environmental
parameters from other COSYNA observations, such as salinity, sea surface temperature, or
chlorophyll facilitates the understanding of the seabirds' foraging behavior, their likely food
intake and habitat choice (Fig. 19). On the other hand, the recorded spatial and temporal flight
patterns and environmental parameters can help to characterize the environmental status of
the North Sea.

## 27   5.9   *In situ* mapping of the COSYNA observation area

The regular operational observations in COSYNA primarily detect variables at the sea surface
(currents observed with HF radar; chlorophyll-a, TSM, and SPMC observed with satellite
remote sensing), at constant depths at fixed high-resolution time-series stations (Wadden Sea



poles, FINO3 platform, MARNET stations), or at constant depth along regular ship routes
(FerryBox transects). In order to observe the vertical distribution of key variables and their
temporal development, these observations were complemented by extended *in situ* mapping
of the North Sea during several research cruises.
In particular, the surveys aimed at investigating the representativeness of single-point time-
series observations, delivering larger-scale validation data for the COSYNA remote sensing
systems and numerical models, testing the functioning of new sensors for permanent missions
under North Sea conditions, and relating concentrations and characteristics of living and non-
living water constituents to optical surrogate variables.
The regular COSYNA mapping grid covers estuarine, Wadden Sea, and open shelf sea water
(Fig. 1). It consists of four East-West and four South-North cross-shore transects and touches
the fixed COSYNA and MARNET stations covering the whole German Exclusive Economic
Zone (EEZ). The land side is limited by a water depth of 10 m and its most seaward reach by
the borders of the German EEZ.
From 2009 to 2013, up to four cruises per year were carried out with RV Heincke. The cruises
took place between March and October to take seasonal variations into consideration. At a
ship's speed of 6 to 8 knots, the grid was completed in less than a week. During this time, the
water masses did not move substantially as confirmed by model studies using Lagrangian
tracers. The observations thus provide a good approximation of the spatial distribution of the
observed variables.
Along the grid lines, an undulating towed Scanfish Mark II ™ by EIVA (Fig. 20) was
operated yielding vertical profiles of oceanographic and bulk biogeochemical parameters at a
vertical resolution of several centimeters and a horizontal resolution of 150 m at mid water
depth. A FerryBox system was used to analyze water continuously taken at a depth of 4 m
with respect to the standard oceanographic parameters temperature, salinity, pH, chlorophyll-
fluorescence, turbidity, CDOM, nutrients, dissolved oxygen, and $pCO_2$. During the cruises,
the FerryBox also served as platform for testing newly developed sensors. This includes a
flow-through PSICAM for high frequency hyperspectral absorption coefficient measurements
(Wollschläger et al. 2013; 2014), a sequential injection analysis (SIA) approach for phosphate
measurement (Frank and Schroeder, 2007), as well as high precision spectrophotometric
methods for the determination of pH and total alkalinity (Aßmann et al. 2011; Aßmann,
2012). Vertical current profiles were recorded with an ADCP. During two cruises, gliders



operated in parallel enhancing the spatial observation density. At the cruise track crossing
points, additional vertical profiles were taken and complemented with Secchi depth
determination, light transmission, and scattering spectra taken from water samples.
As an example, the spatial distribution of $\sigma_T$ (potential density – 1000 kg m$^{-3}$) and
chlorophyll-*a* fluorescence are shown for the cruise at the end of July, 2010 (Fig. 21). Vertical
density gradients at the 5 m thick pycnocline of up to 0.3 kg m$^{-4}$ indicate a strong stratification
typical for the summer months. In the outer reaches of the observation area, two pycnoclines
can be discerned. In the presence of stratification, chlorophyll-*a* shows a typical deep water
maximum at the upper pycnocline. The sudden increase of oxygen saturation directly above
this maximum can be attributed to photosynthesizing phytoplankton. By coupling the
observed vertical distribution of potential density and SPMC with a modeled turbulence
parameter field, the spatial distribution of settling velocities in the COSYNA observation area
was derived (März et al., 2016). Characteristic scales for the coupling of physical
submesoscale and mesoscale processes and the distribution of chlorophyll-a were identified
by North et al. (2016) by applying wavelet analyses to Scanfish data.
**6    Sensor and Instrument Development**
In COSYNA, well-proven commercially available sensors and sensor systems are used.
However, to automatically measure the main parameters that control and influence the North
Sea and arctic ecosystem, several novel, automated, and reliable sensors had to be developed
and tested by the COSYNA partners. These are, in particular, sensors and samplers for
biogeochemical and optical parameters as well as micropollutants. An overview is given in
the following. For most of these sensors, the FerryBox was used as a test platform because it
is protected from the environment, it provides a continuous sea water supply, and offers high-
frequency data acquisition and real-time data transmission.
**6.1    pH Sensor**
pH is a "proxy" for phytoplankton and primary production, one of four parameters
characterizing the oceanic inorganic carbon system, and an indicator for the increasing
acidification of sea water. In order to quantify the components of the carbon cycle in the
context of climate change, a precise characterisation of the carbonate system is required.
In COSYNA, commercially available pH glass electrodes are routinely used.  They are very
sensitive to bio-fouling as bacterial biofilms on the electrodes changes the pH thus requiring




cleaning and re-calibrating intervals of 7-10 d in summer. Although an accuracy of ±0.05-0.1
pH units can be achieved in FerryBox systems for several weeks due to their regular
automatic cleaning procedures, a higher precision of < 0.01 pH units is necessary to detect the
acidification process in coastal waters with a pH decrease of about 0.0019 pH units per year
(Dore et al., 2009; Feely et al., 2009).
In COSYNA, a more precise sensor based on a spectrometric approach was developed
(Aßmann, 2012) that detects the colour of a suitable indicator dye in a miniaturised flow-
through system. A precision of ±0.0007 pH units with an offset of +0.0081 pH units to a
certified standard buffer was be achieved for several weeks. It is, however, not yet suitable for
low-energy applications.
**6.2   Alkalinity Sensor**
$CO_2$ flux estimates for the coastal ocean are subject to large uncertainties (Borges, 2005;
Chen and Borges, 2009) due to strong seasonal variability. For a description of the carbonate
system at least two of the following parameters have to be measured: pH, partial pressure of
$CO_2$ (pCO$_2$), total alkalinity (AT), and total dissolved inorganic carbon (CT). Because a
combination of pH and pCO$_2$ only yields a precision of about 1%, a sensor for the additional
measurement of alkalinity was developed that will allow to document the fast changing
carbonate chemistry in the North Sea (Aßmann, 2012).
The approach for the photometric pH determination (Section 6.1) was modified for alkalinity,
with the advantage that the same equipment can be used for both parameters. The chemical
titration can either be accomplished by using an "open-cell technique" applying a simple sea
water model as calculation tool. The titration occurs at pH <4.5 leading to a removal of all
carbonate species by outgassing of $CO_2$. The precision is ±1.1 mol kg$^{-1}$ with an accuracy of
±8 mol kg$^{-1}$. In a more complex "closed-cell technique" a broader pH range is used and no
$CO_2$ escapes yielding an accuracy of ±0.8 mol kg$^{-1}$ with a precision of ±4.4 mol kg$^{-1}$.
**6.3   Nutrient Sensor**
COSYNA uses commercially available nutrients analysers on FerryBoxes for long term
investigations of the nutrients ammonia, nitrite, nitrate, phosphate, and silicate which are
important parameters regarding eutrophication. However, as small-scale processes often
require faster sensor response times, a flow-through system was developed for the fast



determination of ammonia and phosphate based on sequential injection analysis (SIA) causing
a chemical reaction of both species with a reagent that can be detected by fluorescence (Frank
et al., 2006). The detection limits are 0.3 µmol L$^{-1}$ for phosphate and 1 µmol L$^{-1}$ for ammonia.
180 samples can be processed per hour and analyte.
This reliable analyser is especially useful for high-resolution surface mapping of ammonia
and phosphate in coastal areas and for long-term monitoring due to the low amount of
reagents used in this system (Frank and Schroeder, 2007). Nitrite and nitrate underway
measurements were performed using ultraviolet absorption techniques with parallel
temperature and salinity corrections, thus enabling application of this approach in coastal and
estuarine waters (Zielinski et al., 2011; Frank et al., 2014).
### 6.4  Flow-through Spectral Light Absorption Measurements
One of the most important biogeochemical parameters for the assessment of the
environmental status of the North Sea is the phytoplankton concentration. The standard
method that is routinely used in COSYNA is the continuous *in situ* measurement of
chlorophyll-*a* fluorescence as a proxy for biomass estimation. Since phytoplankton
fluorescence dependents on factors, such as plankton species, plankton physiology, or light
climate, frequent sampling with subsequent lab analysis is necessary to reduce the large errors
of up to one order of magnitude (UNESCO, 1980; SCOR Working Group, 1988).
Better suited to determine estimates of phytoplankton concentrations is the spectral absorption
coefficient. To overcome the disturbing effects of the light scattering of inorganic and organic
suspended matter, a flow-through Point-Source Integrating Cavity Absorption Meter (ft-
PSICAM) was developed in COSYNA yielding continuous measurement of spectral
absorption coefficients in the range of 400–710 nm with high temporal and spatial resolution.
Additional useful information on CDOM/gelbstoff, algal pigments, and suspended matter can
be obtained as well.
By using an integrating sphere, photons cannot get lost and the optical path length is increased
allowing the measurement of very clear waters. This PSICAM principle (Kirk, 1997;
Lerebourg et al., 2002; Röttgers et al., 2005) was modified into a flow-through unit that can
be used unattended on FerryBoxes or other platforms (Wollschläger et al., 2013; Wollschläger
et al., 2014). To reduce the contamination of the integrating sphere, it has to be cleaned
automatically. The ft-PSICAM delivers data with high temporal and spatial resolution.



## 6.5 Molecular Observatory

Information on marine photosynthetic biomass distribution and biogeography with adequate temporal and spatial resolution is needed to better understand consequences of environmental change in marine ecosystems. Since COSYNA methods can only automatically measure proxy parameters for biomass, such as chlorophyll-$a$, a method for the automatic determination of phytoplankton taxonomic composition is required. Molecular analyses, e.g. next generation sequencing (NGS) or molecular sensors are very well suited to provide comprehensive information on marine microbial or protist composition.

In COSYNA, the remotely controlled Automated Filtration System (AUTOFIM) for automated collection of samples for molecular analyses was developed. Resulting samples can either be preserved for later laboratory analyses, or directly subjected to molecular surveillance of key species aboard the ship or at a monitoring side via quantitative polymerase chain reaction or an automated biosensor system (Metfies et al., this issue). The latter is based on an automated pre-treatment of the samples with an ultrasound sample preparation unit that was developed in COSYNA alongside with AUTOFIM. The sampling system can either be deployed on a fixed monitoring platform or aboard a ship for near-real time information on abundance and distribution of phytoplankton key species. Currently, two AUTOFIM-systems are operating on Helgoland and aboard RV Polarstern in order to collect samples for molecular analyses.

## 6.6 Zooplankton Sampling

In addition to phytoplankton distributions, the heterogeneities of the spatio-temporal zooplankton community assemblage are a key environmental parameter. Based on the established Lightframe On-sight Keyspecies Investigation technique (LOKI; Schulz et al., 2010), an imaging head for autonomous, moored operations was developed and attached to the COSYNA Underwater-Node System. A 360°-open flow chamber ensures optimal flow. The data are transferred to shore in near-real time.

LOKI combines several features bringing it close to the feasible borders set by the laws of optics (Schulz, 2013). These are an integrated flash unit providing sufficient light for short shutter times of < 30 μs to avoid motion blurring, very high resolution of <15 μm pixel$^{-1}$ to resolve fine taxonomical characteristics, and a depth of field of several millimetres. This was achieved by using two optical cones (Fig. 22). The first one is attached to the camera housing and allows adjustment of the focal plane at a certain distance from the camera, while the



tapering enhances water exchange in the flow chamber. The opposite cone houses a high-
power LED flash unit. The LEDs are arranged circular and off-axis to provide indirect and
homogenous illumination resulting in high-resolution images of minute specimens and a large
depth of field. The operation time is, however, limited by bio-fouling (Fig. 22).

### 6.7 Active- and Passive Sampling Tools

To determine the potential effects of micropollutants on the marine environment and biota, a
set of integrative active and passive samplers has been developed. Suitable instruments for
unattended use under the harsh conditions do not exist and pure concentration data of
micropollutants are often not very meaningful.
For passive sampling, a Chemcatcher Metal (Petersen et al., 2015b) as well as DGTs have
been used, while blue mussels (*mytilus edulis sp*.) have been applied as active sampling
devices. After a deployment period of several weeks, the samples are analysed with
conventional analytical laboratory methods. In contrast to spot sampling, passive samplers
allow to measure the more representative time weighted average water concentrations (TWA).
Passive sampling data also provide information about the biologically available trace element
fraction of the analysed water body (Booij et al., 2016). Besides the measurement of
contaminant body burdens, the application of mussels as active sampling devices allows also
the analysis of potential biological effects induced by the contaminants present in the
surrounding water. This is done with an analysis of the up and down regulation of specific
proteins, whose expressions are related with certain detoxification mechanisms.
In COSYNA, two systems (Helmholz et al., 2016) have been developed featuring a modular
design for the installation on different instrumental platforms, such as different passive
sampling devices, SPM traps, and cages for biota deployment. An elevator enables the manual
deployment and recovery of the experimental device at a fixed position approximately 3 m
above the sea floor. The use of titanium reduces corrosion. The systems are deployed next to
the FerryBox Station in Cuxhaven at the mouth of the river Elbe and at the MARGate
underwater testing site near Helgoland at a water depth of approximately 10 m.
A continuous flow box has been developed to overcome bio-fouling problems as well as to
minimize effects of changing currents on the sampling rate, as it allows the integration into
FerryBox systems (Petersen et al., 2015b) for passive sampling during ship cruises to obtain
TWA contaminant data.





For the calculation of uptake rates, a calibration was carried out for Ni, Cu, Zn, Cd, Pb, Sc, Ti,
Mn, Co, Ga, Sr, Y, Ba, U and rare earth elements under different environmental conditions
(Petersen et al., 2015a). Up to now, these calibrations were not available for most elements of
environmental concern besides Cu, Cd, Pb, Ni, and Zn. With these developments, a real
multi-element analysis using passive sampling was possible for the first time.

## 6  6.8   Radiometric Ocean Colour Measurements

The colour of the ocean is related to its optically active constituents and can be assessed with
radiometric measurements within the water column and from above the water surface (Moore
et al., 2009; Garaba and Zielinski, 2013a). The latter includes satellite and airborne platforms
as well as measurement poles or vessels (Zielinski et al., 2009).
As part of COSYNA, the applicability of different low altitude hyperspectral radiometer
installations was investigated. Measurement poles at Spiekeroog (Fig. 23) and in the Alfacs
Bay (Ebro Delta, Mediterranean) were outfitted with TriOS RAMSES hyperspectral
radiometers. Underway observations were performed from research vessels Otzum and
Heincke, the latter with a permanent installation of a twin remote sensing reflectance setup to
account for different sun angles along the track.
One of the major challenges is the corruption of data from sun glint and white caps. It is
therefore key for any operational observing system that robust automated quality assurance
methods are applied, which is achieved by parallel image acquisition and analyses (Garaba et
al, 2012) or from spectral feature utilization (Busch et al, 2013; Garaba and Zielinski, 2013b).
An ensemble of sun glint detection methods improves the flagging performance of the data
quality algorithm (Garaba et al., 2015a). The remote sensing spectra of good quality are used
to derive in water constituents like chlorophyll, coloured dissolved organic matter, and
suspended particulate matter along cruise tracks in the North West European Shelf Sea
(Garaba et al., 2014b) and Arctic (Garaba et al., 2013a), and at a time series station in the
Wadden Sea (Garaba et al., 2014a). A very recent application is the calculation of the Forel-
Ule-Colour-Index from reflectance spectra, which opens the possibility to link modern
observations to long term records and to involve citizens with smartphones in ocean colour
measurements (Busch et al., 2016; Garaba et al., 2015b; http://www.eyeonwater.org).



## 6.9 Temperature Sensor for Sediments

To measure the exchange of heat and particulate matter between the German Bight and the Wadden Sea, the heat fluxes between the tidal flats and the water body have to be determined (Onken et al., 2007). As the stratification in the sediment is directly related to the heat content, the latter can easily be calculated and the heat flux between seabed and atmosphere or overlying water derived.

For these investigations, a vertical temperature sediment profiler was developed. The self-contained probe measures the temperature of intertidal sediments between at depths of 0.02 m, 0.1 m, 0.2 m, 0.3 m, and 0. 4 m. Two electrodes located about 2 cm above the sediment indicate whether the tidal flats are wet or dry. The probe was deployed close to the Hörnum measurement pole (Section 5.1.1) where sea water temperatures were measured (Onken et al., 2010).

## 7 Modelling and Data Assimilation

Observations – and even automated observation networks – are limited by the fact that we cannot measure everywhere and at all times, which is in particular a challenge given the coastal ocean's strong variability. One of the distinguishing features of COSYNA lies therefore in the integration of observational data into models in order to close the spatial and temporal gaps of the observations and to calculate energy or matter fluxes. Model studies are also essential for identifying regions with high sensitivity or variability in certain quantities that warrant the deployment of measurement devices. On the other hand, state-of-the-art numerical models of coastal dynamics require monitoring data to reasonably manage large model uncertainties. The observations are used to bring models closer to the "real" state of the ocean, either by verifying model output or by assimilating them into models. These data sets should be representative and coherent. In order to continuously provide accurate pre-operational coastal ocean state estimates and forecasts, COSYNA integrates near-real time measurements in numerical models in a pre-operational way that is meant to improve both historical model runs and forecasts.

In this context, COSYNA has explored different techniques to assimilate data into models. Satisfactory assimilation results were achieved when 2D-data fields were available, such as derived from HF radar or satellite observations. The assimilation of data from single locations or sections usually only influences the immediate vicinity of the locations where the



observations were made and has limited value for greater spatial extensions. Data assimilation
based on physical values is generally more easily achieved than with biogeochemical
quantities. The successful assimilation products of COSYNA encompass surface currents,
significant wave height, period and wave direction, as well as temperature.
For the assimilation of current observations, a nested 3D-hydrodynamic model is used. *In situ*
current time series are measured with stationary ADCPs at the FINO-1 and FINO-3 research
platforms. Remote sensing of surface currents is carried out with three HF radar systems
installed in the German Bight (Section 5.3). For technical details of data processing and
accuracy see Stanev et al. (2015). The flow of observational data including observing nodes,
data management system, and data assimilation capabilities is streamlined toward meeting the
needs for high-quality operational data products in the German Bight (Fig. 24).
Although there are hundreds of HF radar systems installed worldwide, their operational use in
numerical models, in particular at sub-tidal periods, is not well established. The assimilation
of HF radar data is a challenge due to irregular data gaps in time and space, inhomogeneous
observational errors, as well as inconsistencies between boundary forcing and observations.
Furthermore, due to the high sampling frequency of typically several times per hour, it is
difficult for the model to reach equilibrium between two time steps. Therefore, the Spatio-
Temporal Optimal Interpolation (STOI) filter has been developed by Stanev et al. (2015). It
enables a blending of model simulations from a free run and radar observations by extending
the classical Kalman analysis method to time periods of at least one tidal cycle by using the
Kalman analysis equation.
The modelling suite is based on the 3D-primitive equation General Estuarine Transport Model
(GETM; Burchard and Bolding, 2002). It is used in two configurations: a North Sea–Baltic
Sea model of 5.6 km resolution and a one-way nested German Bight model with a horizontal
resolution of about 1 km (Stanev et al., 2011). Both models use terrain-following equidistant
vertical coordinates (s-coordinates) with 21 non-intersecting layers.
The validation of the model and the physical interpretation of the results showed the good
skills of STOI not only in the area covered by HF radar observations but also outside it,
revealing its upscaling capabilities (Stanev et al., 2015). By using HF radar data in the STOI
system, homogeneous and continuous 2D-current fields were thus generated over the entire
model area. The quality is superior to a free model run, demonstrating that data assimilation
can enhance coastal ocean prediction capabilities by making use of observations and





modeling, which is an essential aspect of an operational system. The combination of HF radar
data and numerical model results can therefore also provide a deeper insight into the German
Bight dynamics and provide useful indications where further model developments
(improvements) are needed.
COSYNA also provides a pre-operational wave-forecast based on the WAM Cycle 4 wave
model (release WAM 4.5.3; Komen et al., 1994; Guenther et al., 1992). The computational
system consists of a regional WAM for the North Sea with a spatial resolution of ~5 km and a
nested-grid with a spatial resolution of 900 m for the German Bight. Wind fields and
boundary information are provided by the German Weather Service (DWD) derived from
their regional wave model EWAM. A number of wave parameters such as significant wave
height, period, and total wave direction are calculated (Staneva et al., 2015). It is continuously
providing hindcasts and forecasts since December 2009. Daily at 0:00 UTC and 12:00 UTC, a
24 h regional forecast is issued for the North Sea and a local one for the German Bight. As an
example, a typical wave height distribution with low values close to the coasts and higher
values off shore is shown for the German Bight for 21 April 2010 (Fig. 25).
A combination of biogeochemical observational data and numerical models in COSYNA has
been instrumental for a better understanding of material dynamics including steep cross-shore
gradients ranging from shallow near-shore waters to the continental shelf, strong lateral
gradients and mesoscale patchiness, as well as singular events, such as storms or ice winters.
These processes are intimately linked to the functioning of coastal ecosystems but also affect
efforts to maintain shipping pathways and coastal defense, as well as water quality.
A model- and data-based analysis (März et al., 2016) highlights a remarkable cross-shore
separation of the coastal ocean with a maximum settling velocity of suspended material in the
transition zone between the shallow Wadden Sea and the continental shelf, which modifies
the traditional concept of continuous gradients. This acceleration of vertical deposition fluxes
is likely due to enhanced particle aggregation induced by organic substances, which in turn
are released by planktonic microorganisms (Su et al., 2015; Hofmeister et al., subm.).
Enhanced deposition in the coastal transition zone leads to an effective trapping of lithogenic
material within near-shore waters, while it may act as a barrier for offshore organic particles.
Even higher variability at scales below the cross-shore gradients is evident in COSYNA
lander observations (Section 5.6) of total benthic oxygen consumption.



Using an ecosystem model that includes turbidity fields, estimated from Scanfish observations
(Section 5.9), and accounts for the acclimation capacity of phytoplankton, lateral variability in
chlorophyll-*a* can be reproduced to a remarkable degree (Fig. 26; Wirtz and Kerimoglu,
submitted). However, these reconstructed pelagic patterns decouple from benthic respiration
patterns. Vertical deposition of freshly produced material greatly varies within the coastal
ocean. In a few, mostly deeper regions, deposition prevails over resuspension, leading to
depositional hotspots.
Vertical structures in nutrient concentration are key to understand whether, when, and where
phytoplankton blooms form after storm events (Su et al., 2015). Vertical structures in
chlorophyll-*a* below the meter scale (thin layers) as recently observed by gliders and Scanfish
(Sections 5.2, 5.9) as a persistent feature indicate that a considerable amount of primary
production takes place unnoticed from satellite observations and, as a consequence, also from
many modeling studies. A model validation using COSYNA data thus helps to significantly
improve estimates of total primary production of the German Bight.
**8   Data Management and Data Products**
**8.1   Data Management**
The COSYNA data management system (CODM) was established to make observational and
model data publically available in near-real time (Breitbach et al., 2016). The time between
observations and the availability of data on CODM is ranging from a few minutes for
stationary measurements to about 24 h for data obtained from ships of opportunity and
satellites.
Due to the various observational platforms and model output, it is a significant challenge to
provide a comprehensive overview of the observations with their diverse data formats in
terms of parameters, dimensionality, and observational methods. It is achieved by describing
the data using metadata and by making all data available for different analyses and
visualisations in a combined way independent of data dimensionality. This concerns in
particular the presentation of different data types together in one plot, such as the mapping of
the same variable derived from satellite imagery and *in situ* observations. Key for this is the
harmonisation of parameter names. The various internally used parameter names for the same
observed property are mapped to the corresponding Climate and Forecast (CF) standard name
(Eaton et al., 2010).



Another important aspect of CODM is the use of standardised metadata that are adapted for
the use in direct web service requests (Fig. 27). Two types of metadata are used in CODM:
For observations, the first type describes an observational platform, its sensors, and observed
properties, the second type describes the observed data.
The metadata are created automatically, if the data sets have a distinct beginning and ending.
Examples are ship or glider transects, or single satellite scenes. For stationary platforms, only
one metadata record is created for the entire time-series. For models, the first type of metadata
describes the model itself, while the second type is describing the model run. Data-metadata
are ISO19115 and INSPIRE compliant (EC Directive, 2007) and contain all necessary
information to access the data as download, plot, or map. The metadata itself are also mapped
to a Web Feature Service (Fig. 28).
The observational data have to pass a number of automated and supervised tests, before they
become publically accessible in the data portal. Depending on the test results for range, stuck
values, spikes, and – for some parameters – gradients quality flags are assigned to the data.
The procedures and quality flags are in line with international guidelines (Breitbach et al.,
2016; SeaDataNet, 2010).
CODM is a publically available Open Data portal. There are no restrictions or fees for
downloading and using the data, but CODM requires a basic user registration. Users are asked
to provide country of origin, a user category, and the city. No other personal information is
mandatory. Users are also asked to acknowledge COSYNA as data source in their
publications. The majority of users are in the science sector followed by administration (Fig.

22   29).

## 8.2   Data Products
COSYNA is monitoring the current state of the coastal system in the North Sea and is
generating modelled pre-operational state reconstructions and forecasts. These routinely
provided data can be grouped into four "product" categories:
a) High-resolution time series at fixed positions: Meteorological, oceanographic, water
quality, and biological parameters are continuously observed at the measuring poles (Section
5.1) Spiekeroog, Hörnum Deep, and Elbe, the research platform FINO3 (Section 5.1.2), and at
the stationary FerryBox systems in Cuxhaven and Helgoland (Section 5.4).



b) Repeated transects: Oceanographic and biogeochemical parameters are measured during regular ship and glider surveys (Sections 5.2, 5.9) and with automated FerryBox systems on ships of opportunity (Section 5.4).

c) Remote sensing information: Regular maps of currents, chlorophyll distribution and optical sea water properties are obtained with remote sensing by HF radar (Section 5.3) and satellites (Section 5.7). The data cover large areas of the German Bight and are integrated with observational in-situ data.

d) Integrated COSYNA products: The automatically produced data fields of the German Bight are continuous in space and time and provide hindcast, nowcast, and short-term forecasts. The latter two are improved with data assimilation procedures (Section 7).

The COSYNA product "Surface Current Fields" provides data fields and maps of tidal hindcasts and forecasts of sea surface currents in the German Bight. The fields are updated every 30 min. They are created by assimilating regular HF radar measurements into a 3D circulation model (Stanev et al., 2011; 2015; Section 7).

The pre-operational COSYNA wave forecast model system runs twice a day and provides a 72 h forecast on the regional scale for the North Sea and on the local scale for the German Bight. Significant wave height, period, and total wave direction are calculated (Staneva et al., 2014).

In order to provide the spatial distribution of sea surface temperature and salinity in the North Sea, FerryBox observations taken along ship tracks are extrapolated to larger areas combining them with information from numerical models. Data from the route Cuxhaven-Immingham are assimilated into a three-dimensional circulation model every 24 h (Grayek et al., 2011).

## 9    Outreach and Stakeholder Interaction

COSYNA aims to make scientific data, results, and data products publically available by reaching out to different target groups and users, such as the scientific community, potential users in business enterprises and authorities, and to the general public. To serve this purpose, COSYNA publishes several print products in German and English that are publically available for download at the COSYNA website, or can be ordered. Flyers and more comprehensive brochures provide an overview of the goals, approaches, activities, and results of COSYNA. The annual progress reports are intended for COSYNA partners and users and describe selected results and activities of the various working groups and subprojects within


COSYNA. Newsletter and product fact sheets provide COSYNA partners and users as well as
interest groups or the general public with information on activities, events, or data products.
COSYNA maintains the website www.cosyna.de that informs about motivation, approach,
observations, modelling, products, and outreach activities. The COSYNA data portal is linked
to that web site and provides access to data download and visualisation. On average, the
COSYNA website has been visited by more than 500 different external visitors per month.
Furthermore, COSYNA has developed an interactive app with versions for iPad and other
tablet PCs as well as Android and iOS based smartphones. The app provides explanatory texts
and pictures describing the observing systems, instruments, models and products, as well as
the COSYNA partners. Near real-time data for several platforms are available. COSYNA is
also presenting the app in permanent exhibits in museums, or temporarily at public events or
trade shows.
It is one of the main goals of COSYNA to bridge the gap between operational oceanography
and the users of marine data in local authorities, non-governmental organizations, science and
industry. In order to ensure that products are applicable, COSYNA has been initiating a
dialogue with stakeholders allowing for direct feedback and input to COSYNA. In the initial
phase of COSYNA, a national and an international survey showed that the COSYNA data
products are useful to a great number of users from different sectors and fit into the
international context. Follow-up workshops and an external evaluation of the integrated
COSYNA product "Surface Current Fields" have clearly improved COSYNA products and
their usability. To explore the streamlining of COSYNA products for the offshore wind
energy industry, several workshops were held to pave the way for future co-operation with
offshore wind energy companies (Eschenbach, this issue).

## 10  Conclusions and Outlook

COSYNA was established with its sight on understanding the state and variability of complex
interdisciplinary processes in the North Sea and the Arctic. During its first years, work
concentrated on establishing the observational network, developing sensors and numerical
models, testing and applying data assimilation techniques, building a data management
system and testing outreach strategies. Now, that the core of what had been envisioned in the
original concepts is operational and functioning, COSYNA will expand into new areas,
spatially as well as scientifically.





Currently, COSYNA is being extended to the western part of the Baltic Sea (in cooperation
with a new partner, GEOMAR, Helmholtz Centre for Ocean Research) by installing an
Underwater-Node System in spring 2016 in the Eckernförde Bight near the location of
GEOMAR's long established Boknis Eck time-series station (Lennartz et al., 2014).
COSYNA already contributes to observations of other coastal areas in the world, such as the
Lena delta, the Bohai Sea in China, or with instruments on research vessels and cruise ships
operating in various parts of the world ocean. In the long run, COSYNA will be part of
HZG's Global Coast project that aims at identifying representative coastal regions worldwide
that will help evaluate the role of coastal areas for global processes, while using a global
context for understanding regional and coastal processes.
To this end and for use in large national and international research projects, COSYNA plans
to develop mobile observing systems with high resolution capabilities in space and time, that
have very short deployment times in order to be able to react to extreme events such as storms
and floods. As the focus of research projects will be shifting more and more to an integrated
understanding of complex systems, this approach will require cooperation with partners in the
atmospheric and terrestrial research communities. In the future, COSYNA will be closely
interlinked with the Elbe River Supersite of DANUBIUS, the most recent European ESFRI
Roadmap project studying river-delta-sea systems, and will be part of the Helmholtz
Association's MOSES (Modular Observing System for the Earth System) research
infrastructure.
Intensified modeling efforts, especially regarding biogeochemical models and data
assimilation are needed to put the COSYNA observations in a broad context and help
understand coastal systems. This will also yield future data products including wind fields,
ship detection, and biogeochemical parameters. Chlorophyll maps and maps of suspended
particulate matter will be obtained from satellites on a regular basis. The assimilation of other
quantities is work in progress and will be published, when they become available.
The successful technology development of underwater nodes will continue. Currently,
experiments with smaller, more flexible units are underway. Alternative forms of power
supplies, such as fuel cells, are being tested and may allow for a flexible network of nodes.
New partners are joining COSYNA: GEOMAR in Kiel and the Franzius-Institute for
Hydraulic, Estuarine, and Coastal Engineering at the University of Hannover have recently
agreed to become COSYNA partners. For the future, discussions with international partner



will be sought and international cooperation will be intensified – in particular with the
countries bordering the North Sea.
While COSYNA has evolved into a well-established integrated pre-operational observing
system, research will become more central to defining COSYNA's endeavors. Utilizing the
combined expertise of its various partner institutions, COSYNA's science foci will include
biogeochemical cycles from rivers to the North Sea and the Northern Atlantic, the role of
wind farms for physical, biogeochemical, and biological processes in the coastal ocean as
well as associated engineering questions, Land – Wadden Sea – North Sea exchange
processes with an extensive experiment spanning from the Netherlands, along the German
coast to Denmark involving physics and biogeochemistry, and exploration of the possibilities
and challenges associated with citizen science.
**Acknowledgements**
COSYNA was implemented between 2010 and 2014. The infrastructure and instrumentation
of COSYNA was funded by the German Federal Ministry of Education and Research through
the Helmholtz Association. COSYNA is developed and operated jointly between 11 partner
institutions entirely contributing personnel and funding for development, operation, and
maintenance. COSYNA is coordinated by the Helmholtz-Zentrum Geesthacht. Currently,
funding is available through the Advanced Remote Sensing – Ground Truth Demo and Test
Facilities (ACROSS) infrastructure of the Helmholtz Association.
The implementation and operation of COSYNA would not have been possible without the
competent and tireless efforts of the technical staff at all involved institution. We thank the
master and the crew of RV Heincke for their help and support during the ship cruises. The
cruises were conducted under the grant numbers AWI_HE298_00, AWI_HE303_00,
AWI_HE308_00, AWI_HE312_00, AWI_HE319_00, AWI_HE325_00, AWI_HE331_00,
AWI_HE336_00, AWI_HE353_00, AWI_HE359_00, AWI_HE365_00, AWI_HE371_00,
AWI_HE391_00, AWI_HE397_00, AWI_HE407_00, AWI_HE412_00 AWI_HE417_00,
AWI_HE441_00, and AWI_HE447_00.

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



1    Table 1. Standard COSYNA observables.

| Platform | Parameter |
| --- | --- |
| Meteorology | pressure, temperature, global radiation, wind vector |
| Physical oceanography | pressure, temperature, salinity, current vector, wave height, and direction |
| Biogeochemistry | optical turbidity, total suspended matter concentration, chlorophyll-a concentration, oxygen |

3    Table 2. Fixed platforms used in COSYNA. Abbreviations: M: meteorology, P: physical
4    oceanography, B: biogeochemistry. For abbreviations of the partner institutions see Section 1.

| Platform | Years | Position | Mean tidal range [m] | Parameters | Partners |
| --- | --- | --- | --- | --- | --- |
| Pole Hörnum Deep | 2002-2013 (Mar-Nov) | 54°47.6'N 008°27.1'E | 2.3 | M, P, B | HZG |
| Pole Elbe Estuary | 2012-2013 (Mar-Nov) | 53°51.5'N 008°56.6'E | 2.8 | M, P, B | HPA, HZG |
| Pole Spiekeroog | 2002-now (year round) | 53°45.0'N 007° 40.3'E | 2.8 | M, P, B | ICBM |
| FerryBox FINO-3 | 2011-2016 (year round) | 55°11,7'N 007° 9,5'E | 0.9 | P, B | HZG |
| FerryBox Cuxhaven | 2010-now (year round) | 53°52,6'N 008° 42,3'E | 2.9 | P, B | HZG |
| Lander | | n.a. | n.a. | P, B | MARUM, AWI, HZG |
| Underwater Node Helgoland | 2012 – now (year-round) | 59° 11'N 008°52,8'E | | P, B | AWI, HZG |
| Underwater Node Spitsbergen | 2012 – now (year-round) | 78° 92'N, 011° 9'E | | P, B | AWI, HZG |





Table 3. Moving Platforms used in COSYNA. Time resolution is given between repeated measurements at the same location. Abbreviations: M: meteorology, P: physical oceanography, B: biogeochemistry; S: water surface, U: upper water column, FC: Full water column. The abbreviations of the partner institutions are explained in Section 1.

| Platform | Vertical range | Time resolution | Parameters | Partner |
|---|---|---|---|---|
| FerryBox | U | ½ day to a week | P, B | HZG |
| Glider | FC | days to months | P, B | HZG |
| Seabird | U | - | P | FTZ |
| HF radar | S | 20 min. | P | HZG |
| Satellites | S | 2 times in 3 days | B | HZG |
| Ship surveys | FC | months | M, P, B | HZG |



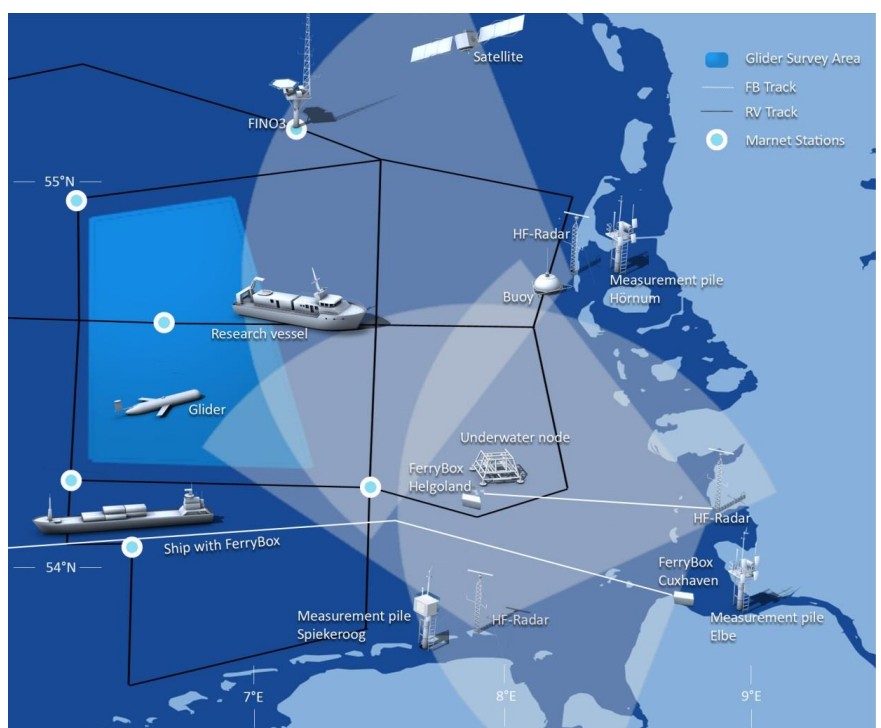

Fig. 1. Map of the German Bight of the North Sea showing the pre-operational components of
the coastal observing system COSYNA.

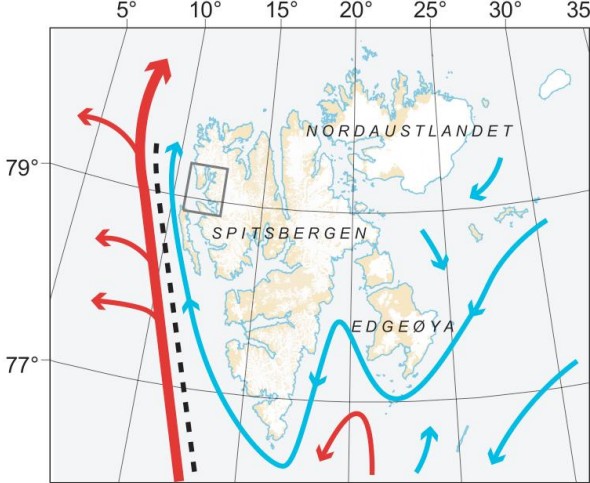

Fig. 2. Spitsbergen with Kongsfjord (small rectangle) at the west coast of Svalbard. Arrows
indicate the warmer Atlantic water masses (red) from the West Spitsbergen current and by
colder less saline Arctic water (blue) from the East Spitsbergen (Cottier et al., 2005).

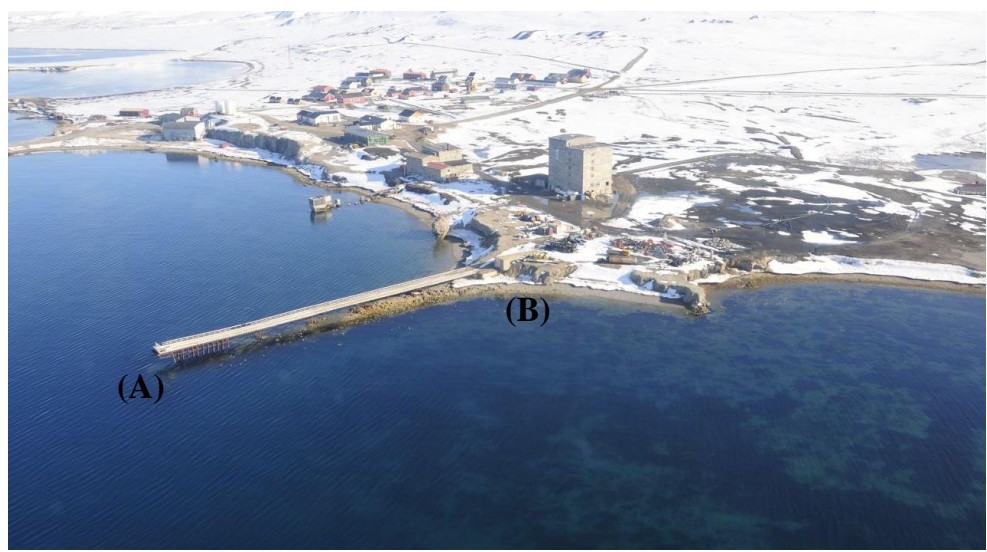

Fig. 3. Research village NyÅlesund. The Spitsbergen Underwater-Node is located about 30 m
in front of the "Old Pier" (A). The control station is located at the base of the old pier on land
(B).

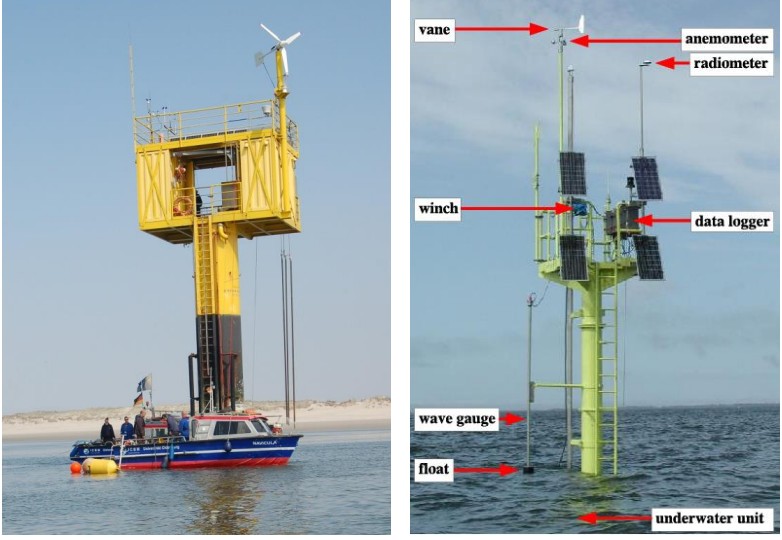

Fig. 4. The measuring poles at Spiekeroog and in the inner Hörnum tidal basin.



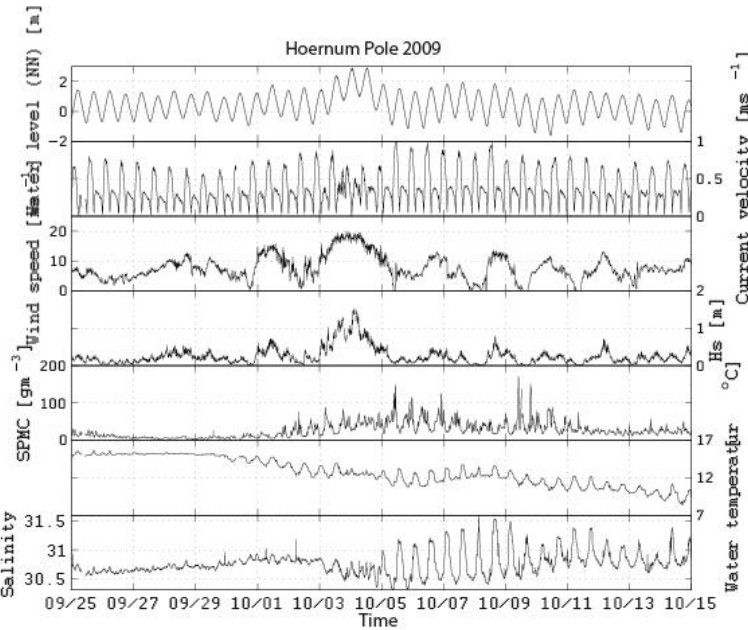

2    Fig. 5. Time-series of the measuring pole in the Hörnum tidal basin showing one week of data

3    with a sampling frequency of 10 min.




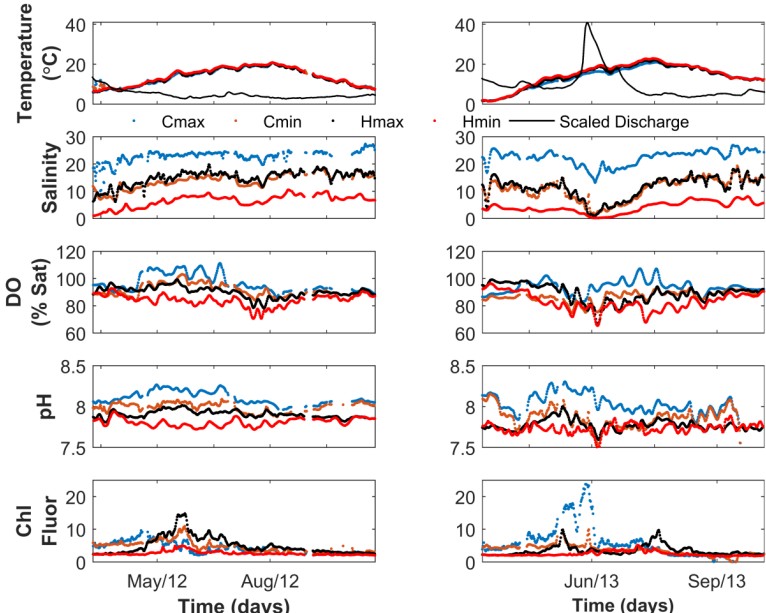

Fig. 6. Time series of the stationary FerryBox located at Cuxhaven at the Elbe river mouth for
2012 (left panels) and 2013 (right panels). Top to bottom: water temperature and Elbe river
discharge (m$^3$ s$^{-1}$) at Neu Darchau station scaled by dividing it by 100 (thin black line),
salinity, dissolved oxygen saturation, pH, chlorophyll-a fluorescence. Shown are the
Cuxhaven values at low tide (brown, Cmin), high tide (blue, Cmax) and from the Elbe estuary
measurement pole at low tide (red, Hmin) and high tide (black, Hmax).

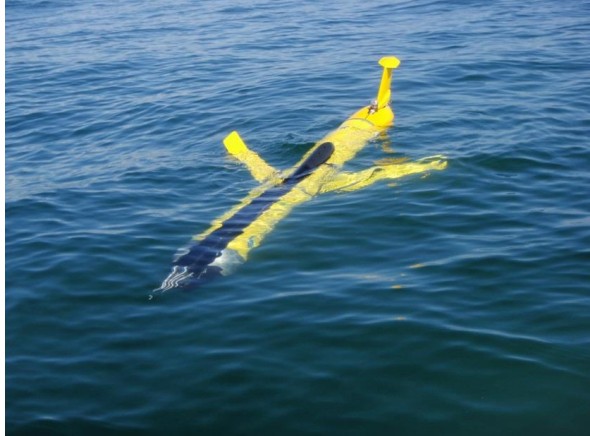

Fig. 7. Ocean Glider surfacing to submit data to shore. The glider is equipped with CTD,
optical sensors, and an additional turbulence sensor.



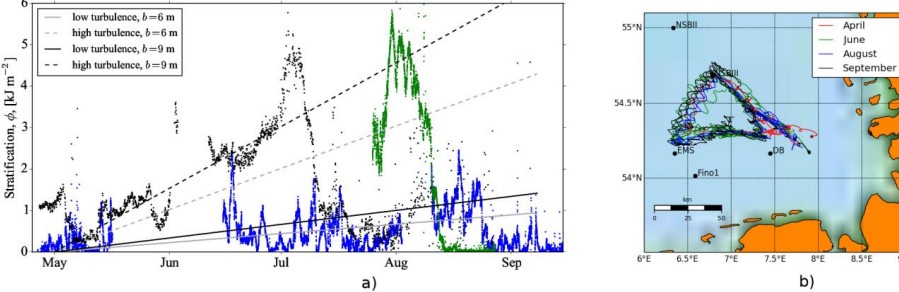

Fig. 8. Left panel: Glider observations showing the build-up of stratification in 2012 and
2014. The stratification is expressed as the energy required to fully mix the water column. It is
computed as $\theta(t) = \int_0^H [\rho_{mix} - \rho(z, t)]gz\, dz$, where $H$ is the water depth, $\rho$ the density, $g$ the
gravitational acceleration, and $z$ and $t$ the vertical coordinate and time (Carpenter et al., 2016).
Right panel: glider tracks in 2012.

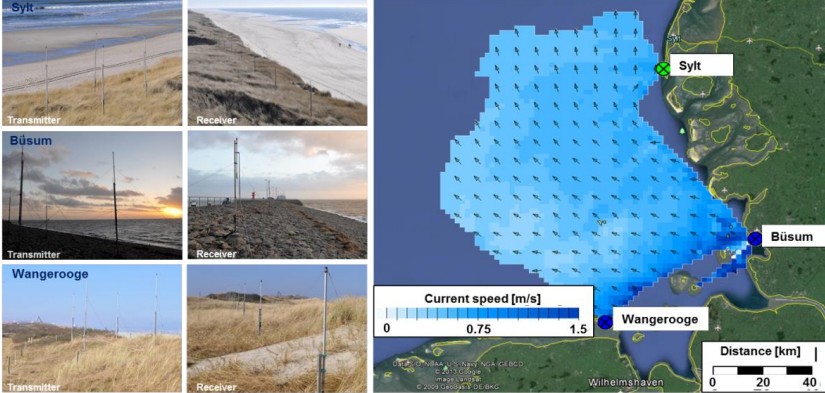

Fig. 9. HF radar system in the German Bight with its three stations in Büsum and on the isles
of Sylt and Wangerooge. The right panel shows an example of the 2D-current field derived
from overlapping radar signals.



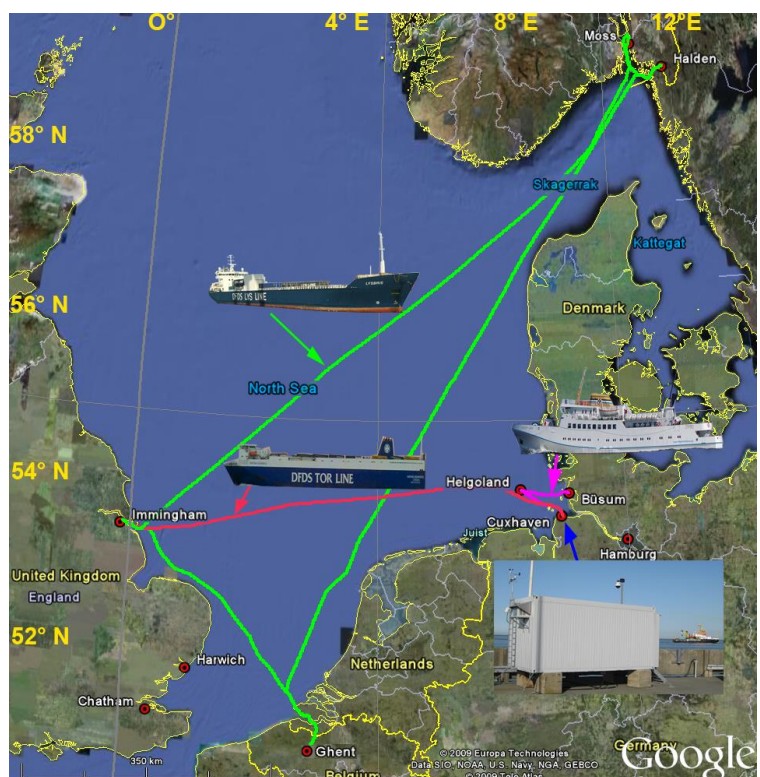

Fig. 10. Map of FerryBox routes and stationary platforms equipped with FerryBoxes.

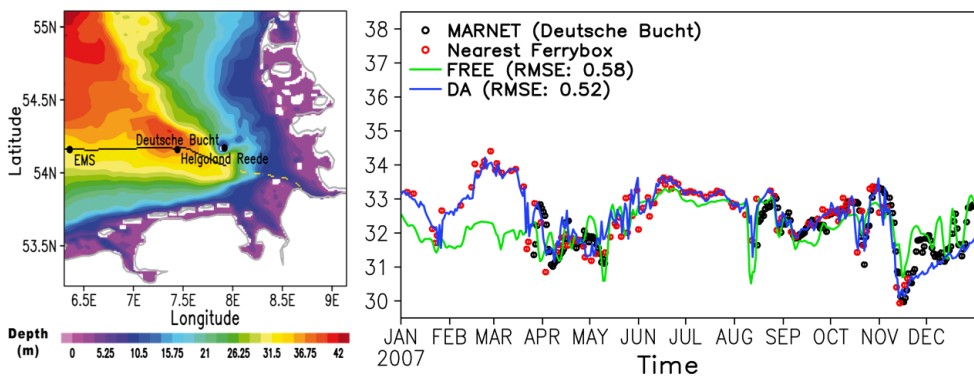

Fig. 11. Left panel: Topography of German Bight and FerryBox track. Right panel:
Comparison of simulated sea surface temperature from a free model run and a run with data
assimilation (DA) against MARNET and nearest FerryBox observations (Grayek et al., 2011).



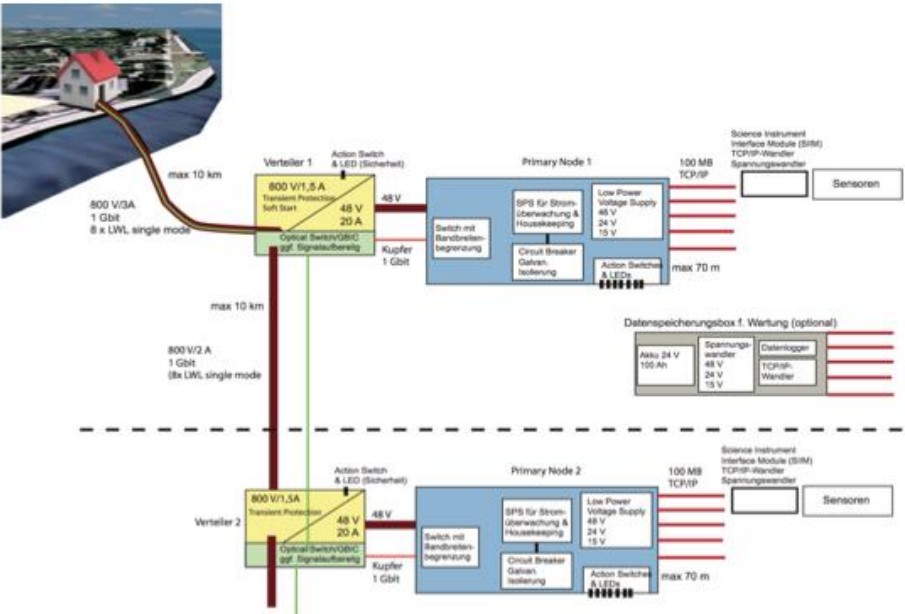

Fig. 12. Setup of the COSYNA Underwater-Node System with (1) land-based server and
power supply, (2) cable connection (max. 10 km) to the first primary underwater node, (3)
primary node, (4) sensors attached to first node, (5) cable connection (max. 10 km) to a
second underwater node, and (6) second underwater node. A third node can be connected.

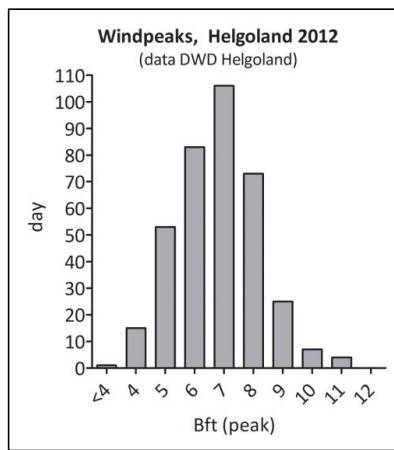

Fig. 13. Number of days per year with windpeaks above a certain wind force at the island of
Helgoland (Source: Deutscher Wetterdienst).



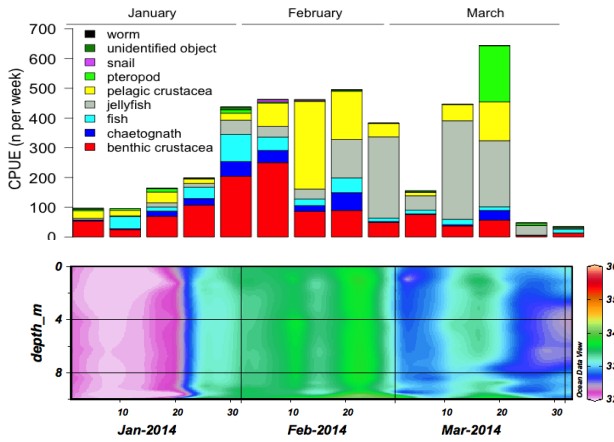

Fig. 14. Upper panel: The temporal abundances of the main biota groups assessed with a
stereo-optic sensor attached to the Underwater-Node System in Spitsbergen. Lower panel: the
temporal pattern of the temperature during the same time period measured with a vertical
profiling CTD at the underwater node system.

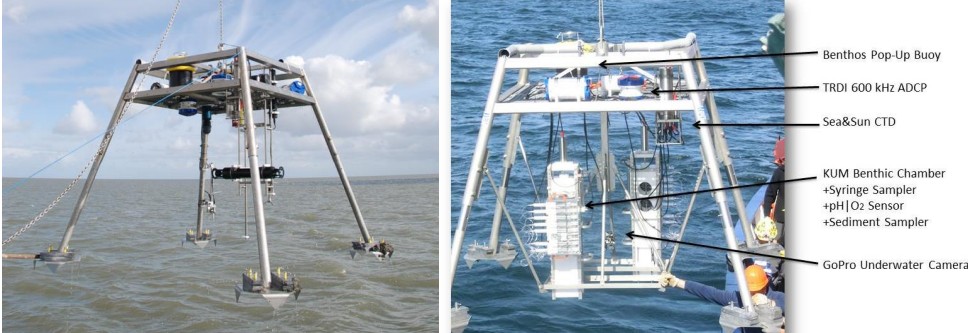

Fig. 15. Deployment of landers a) SedObs (Photo by C. Walcher, AWI) and b) NuSObs.

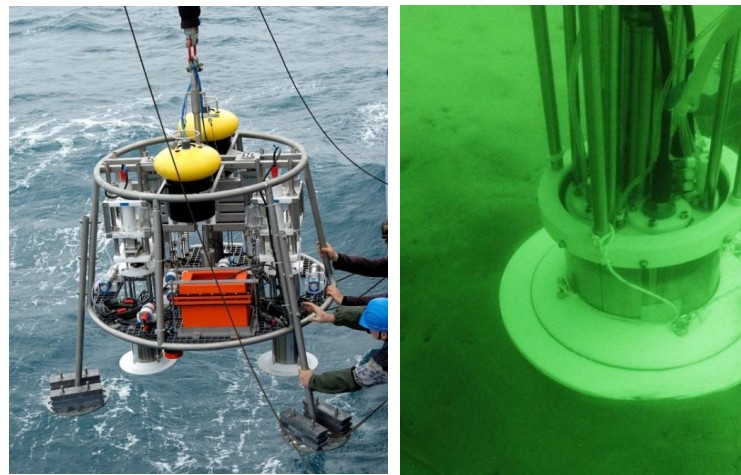

Fig. 16. Left panel: lander FLUXSO deployed for autonomous sampling in June 2015; right
panel: sampling chambers in mobile fine sand at 25 m depth.

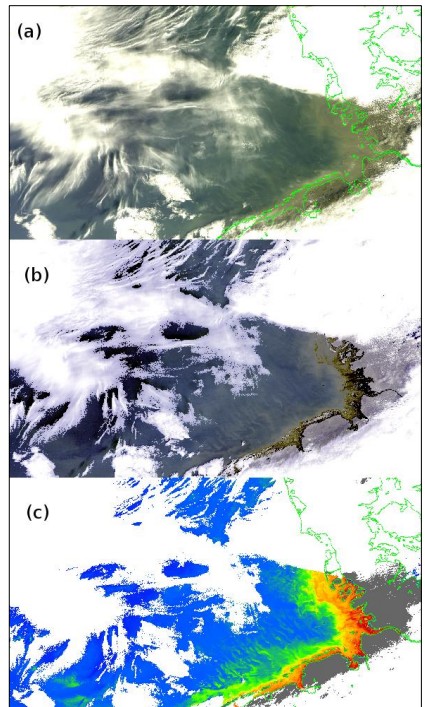

Fig. 17. Satellite scene of the German Bight taken on 2012-03-10 by MERIS. a) Radiance in
atmosphere; b) reflectance at the bottom of the atmosphere (after atmospheric correction); c)
chlorophyll concentration showing filaments of *phaeocystis* blooms along the west- and east-
Frisian coast.





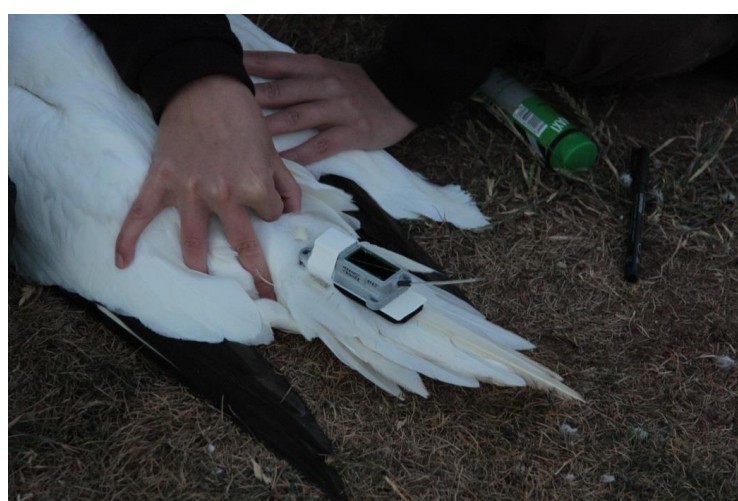

Fig. 18. Solar-powered GPS data logger attached to a tail of a Northern Gannet (Photo: J.
Dierschke).

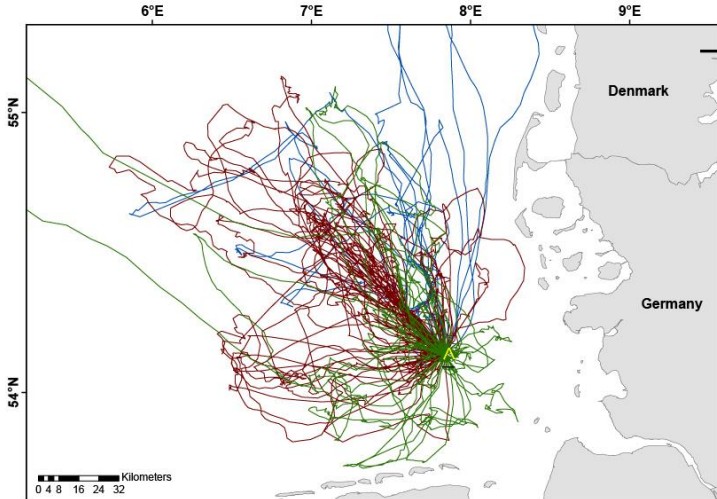

Fig. 19. Foraging flights of three Northern Gannets (*Morus bassanus*) in 2015 starting from
Helgoland.





2    Fig. 20. Scanfish used on regular ship surveys.

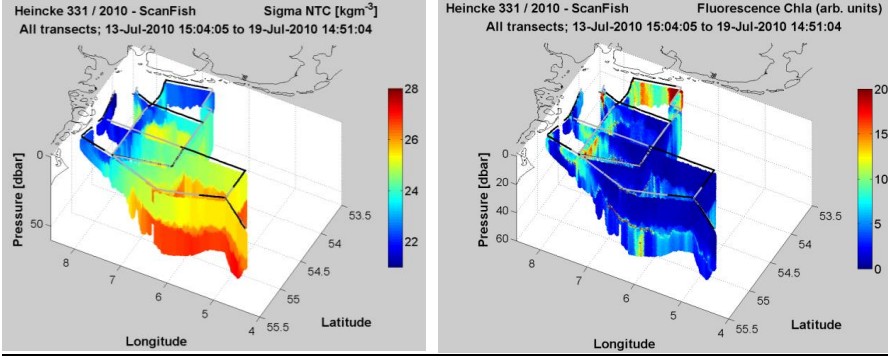

5    Fig. 21. Spatial distribution of $\sigma_T$ and chlorphyll-*a* observed during RV Heincke Cruise
6    HE331 in July 2010.



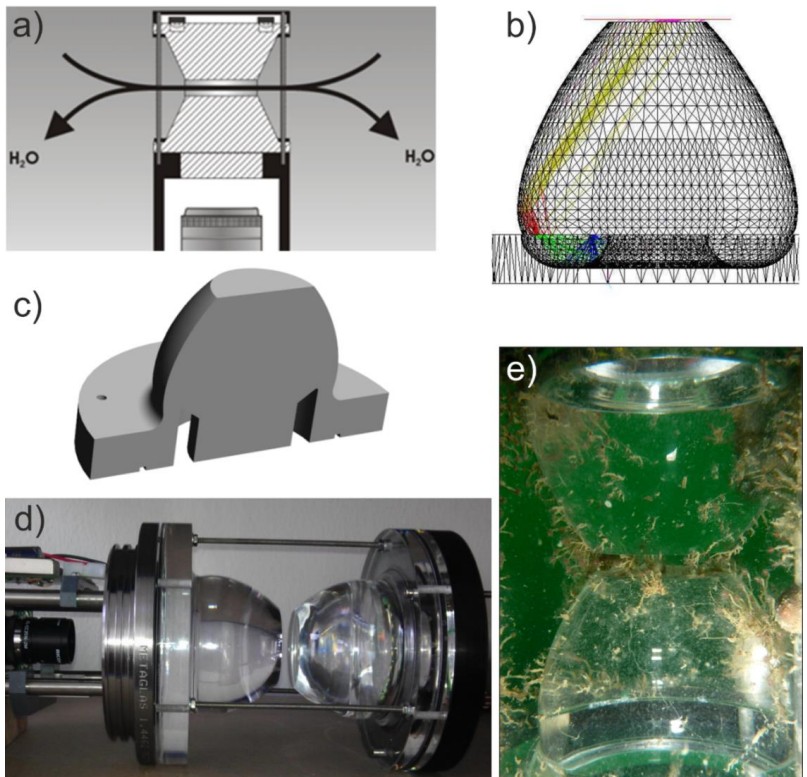

Fig. 22. Design of the LOKI imaging head for moored operation. a) Schematic overview. b)
Ray-tracing design-model to investigate the best shape to increase efficiency. c) Cross-section
of a 3D-model. The LEDs of the flash unit are positioned in the notch. d) Imaging head with
two optical cones: the right cone carries the circular flash unit, the left one the visual path of
the camera's field of view. The camera is mounted on the left. e) The system requires
periodical cleaning in the field. The image shows bio-fouling after 5 weeks of operation in the
North Sea.

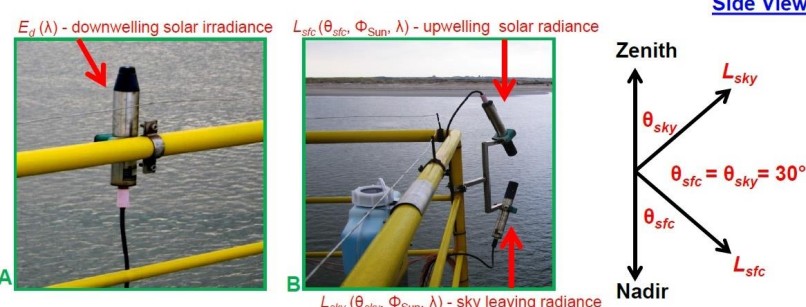

Fig. 23. Setup of RAMSES radiometers at the Wadden Sea measurement pole Spiekeroog.



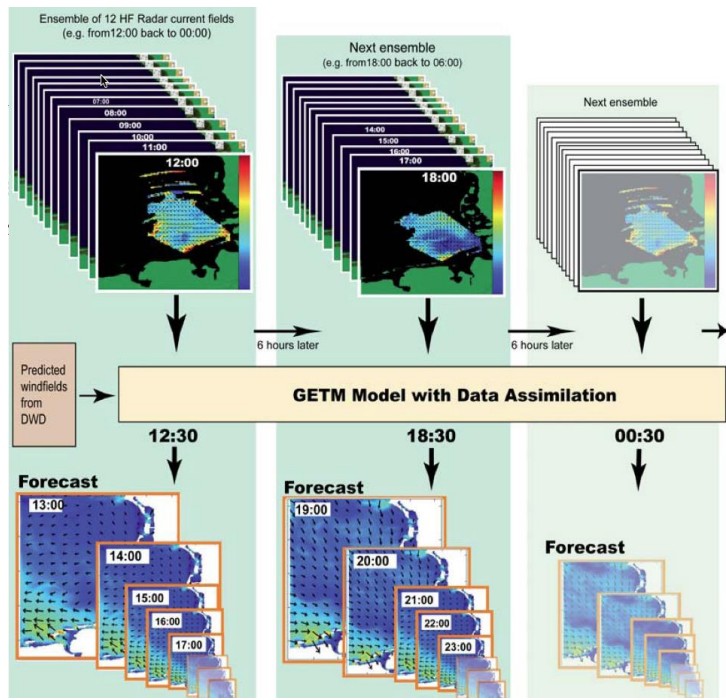

Fig. 24. The functioning of data assimilation and forecasting in the pre-operational COSYNA
system. HF radar system covering the German Bight. Radial current components are sent to
the HZG data server, where current vectors are calculated and presented on the COSYNA
data portal (Stanev et al., 2015).

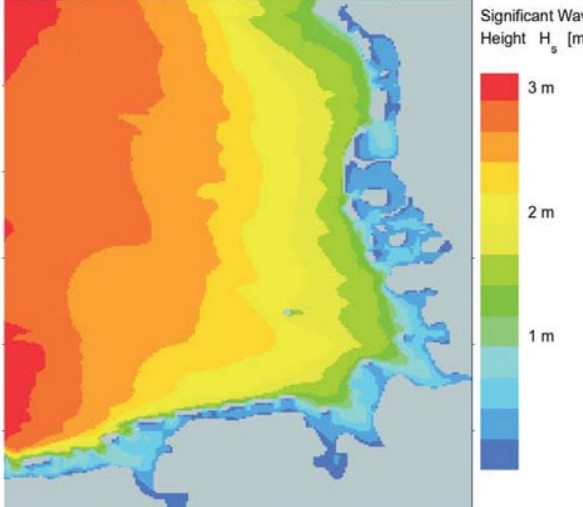

Fig. 25. Significant wave height calculated for the German Bay on 21 April 2010 with the
WAM wave model used in COSYNA.




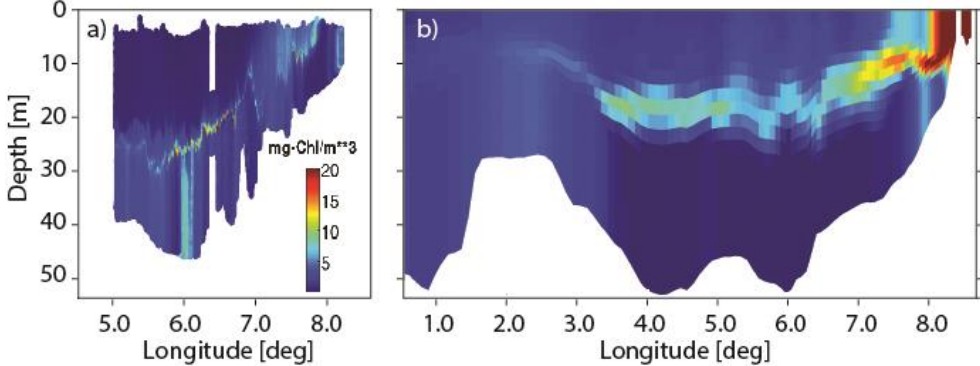

Fig. 26. Chlorophyll transects around 55°15' latitude in the German Bight
a) observed with a Scanfish in July 2010 (Section 5.9) and b) as result
of a coupled GETM and an adaptive ecosystem model showing a 1-week mean
(Wirtz and Kerimoglu, submitted).

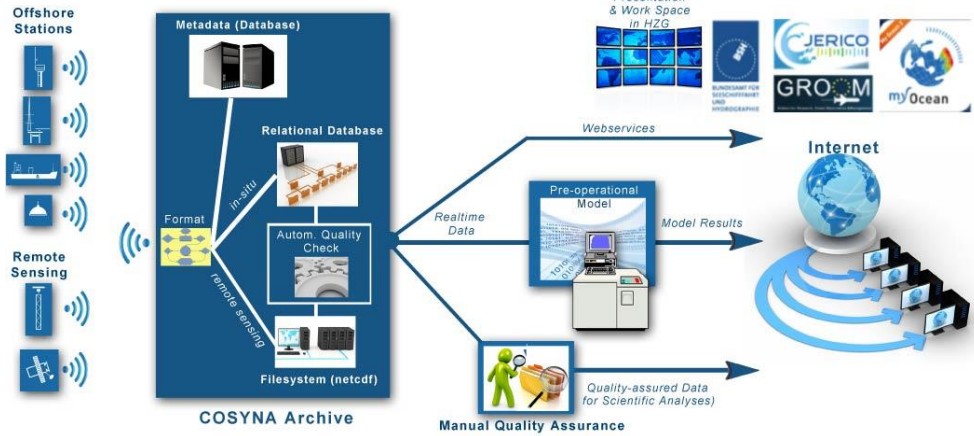

Fig. 27: Data Flow in COSYNA.




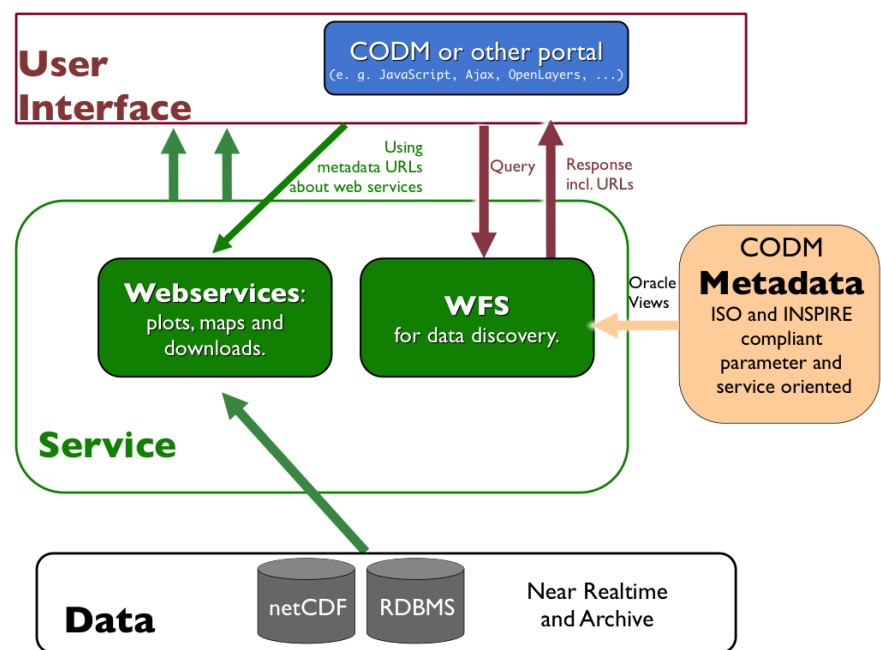

Fig. 28. Data Management architecture: The connection between user interface on one side
and data or metadata on the other side is handled solely by web services like Web Feature
Services (WFS) or Web Map Services (WMS).

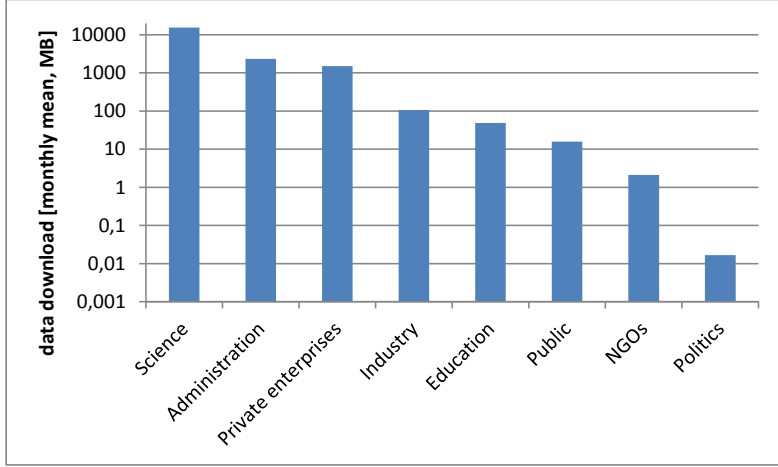

Fig. 29. Mean monthly data use for different categories of users. Data are shown for the time
period between November 2014, when the user registration started, and January 2016.