# Peer review of "The Coastal Observing System for Northern and Arctic Seas (COSYNA)"

_Ocean Science, 2016_

## Referee Comment (RC1) · Anonymous Referee #1 · 27 Sep 2016

\*\*\* General comments

The manuscript proposed by B. Baschek et al. aims giving a detailed overview of the COSYNA, an integrated pre-operational observing system in Northern and Arctic Seas.

As we understand, the article introduces the Special Issue dedicated to COSYNA in Ocean Science. Following this aim, no specific scientific question is addressed in the manuscript but it is more dedicated to the description of the COSYNA components.

As a main general comment, the description is too long and confusing to highlight and to describe the successful integration operated in COSYNA. The balance between the

amount of details and sometimes missing information or imprecisions is harming the main idea. A suggestion could be to shorten the manuscript to emphasize more the strength and coherence of the integrated system.

The description remains also difficult to follow for non-expert readers from the geographical region. A lot of places are mentioned without an illustration on a map. The manuscript will strongly benefit from a general map with zooms and mentioned names in the text (for example: Otzumer Balje, Jade Bay, island of Helgoland, North Frisian, Lena delta, Weser, Ems).

A final general remark is more on the "network" strategy behind COSYNA. The scientific aims and the justifications of geographical extent do not appear clearly from the manuscript.

Considering the needed improvements included in general and specific comments, I recommend this paper for potential publication after minor revisions more related to the shape of the paper.

*** Specific comments
* Abstract
p. 2 / l. 2-3 - Authors mention that COSYNA is designed also to "assess the impact of anthropogenically induced change". This assessment is not directly provided by the network but after complex scientific analysis of the collected data. The sentence should be modified.

* 1. Introduction
p. 3 / l. 28-32 and p.4 / l. 1-7 - The list of contributors could be presented as a table to

improve the sentence.

p. 4 / l. 9 - The paper does not introduce scientific studies but just illustrate the observation collected.

p. 4 / l. 10 - The "volume", I guess the Special Issue is just mentioned here. It should be mentioned before and more explicitly.

p. 4 / l. 11-15 - The structure of the paper needs to be given more explicitly. At least, different main parts must be linked to the section numbers.

* 2. Coastal focus regions
General comment: This section should be strongly reduced. A map could give an overview of the two regions. Some details in this context are not useful as, for example:
- UNESCO reference - p. 5 / l.4-5
- p. 5 / l.21-26
- p. 6 / l.4-8
- p. 7 / l.4-8

p. 5 / l. 6 - The North Sea should be describe before the German Bight.

* 3. Objectives and Benefits
p. 8 / l. 14-15 - I don't see how numerical model can integrate observations from turbulence (and from minutes) as it is scales which are not taken into account in assimilation (observations are generally degraded to be assimilated). This sentence must be clarified.

* 4. International context
p. 9 / l. 20-27 - These statements are not necessary for the manuscript.

* 5. Observations
p. 10 / l. 23 - What is the meaning of "(3)" and "(1)" ? Same for "(1)" and "(2)" in p. 11 / l. 8

p. 11 / l. 10-11 - The mention to CDOM, yellow substances, "gelbstoff" should homogenize in the manuscript. I would suggest keeping Coloured Dissolved Organic Matter (CDOM) as it is a name more commonly used.

p. 11 / l. 30 - For the quality control processes, do you have references or standards to refer ?

p. 16 / l. 29-30 - Could the authors explain more clearly the "order of 10m" ?

p. 17 / l. 8 - The notion of "fusion" is not straightforward or I do not understand what is meant here. Authors should detail a bit more here.

p. 18 / l. 2-3 - The statement is already said before in the manuscript.

5.5 Underwater-Node System - It is not clear to me. What is the depth of the system ?

p. 21 / l. 13 - Typical deployments times exceed 25 h but until how much time, it can

be extended ?

* 6. Sensor and Instrument Development
p. 26 / l. 26 - pH is not only a proxy for phytoplankton and primary production as the water pH is not only driven (even if largely influenced) by biology.

* 7. Modelling and Data Assimilation
Are FerryBox data assimilated in COSYNA modelling system ?

What are the forecast periods for hydrodynamics ? It is detailed for waves but not for transport.

p. 35 / l. 4 - I do not agree that it is "reproduced to a remarkable degree". The model is able to reproduce a deep chlorophyll maximum but its intensity and extent is not similar with observations.

*** Minor and technical corrections
* 2. Coastal focus regions
p. 4 / l.25 - 26 - Sentences could be rephrased ... "It is .... It is ....".

p. 6 / l.24-25 - "a-1" needs to be placed by "y-1".

* 3. Objectives and Benefits
p. 8 / l. 30 - "radar" has to replaced by "HF radar".

* 4. International Context
p. 9 / l. 31-32 - "to providing ... and carrying" to be replaced by "to provide ... and carry"

* 5. Observations
p. 11 / l. 30 - CODM website could be given.

p. 12 / l. 2 - The number (4 or 5) of fixed stations can be explicitly written.

p. 12 / l. 22-23 - Please homogenize the writing. In this part we find back past verb when present is used in other parts.

p. 13 / l. 4-6 - The maintenance frequency could be given.

p. 16 / l. 14 - "The measurements taken with COSYNA gliders is ...." => " ... gliders are ...".

p. 16 / l. 17 - "The data was ..." => "The data were ...".

p. 17 / l. 1 - "The Systems ..." => "The systems ...".

p. 18 / l. 25 - "und ..." => "and ...".

p. 21 / l. 15 - The acrnomy ADCP should be mentioned in brackets before to be used.

p. 21 / l. 21 + p. 22 / l. 14 - The way of writing in situ (in situ, in-situ) must be homogeneous in the manuscript.

p. 23 / l. 21 - "chlorophyll"-a concentration.

p. 24 / l. 29 - "chlorophyll-a" => "chlorophyll-a concentration".

p. 25 / l. 28 - PSICAM is not defined at this stage in the manuscript.

* 6. Sensor and Instrument Development
p. 28 / l. 15-16 - "phytoplankton fluorescence" => "fluorescence".

p. 28 / l. 16 - "dependents" => "depends".

p. 30 / l. 4 - "Fig. 22" => "Fig. 22e".

* Tables
Table 1:
+ "current vector" => "current" or "current velocity"
+ "oxygen" => "dissolved oxygen"

Table 2:
+ the level/depth of measurement is missing.
+ For FINO-3, is it 2016 of now ?

* Figures
General comment: There is a gap between this manuscript and figures ready for

publication because they are often blurred and some are even not readable.

+ Figure 1 is blurred in the pdf.

+ Figure 4, reference (left/right) to the pictures is missing in the caption.

+ Figure 5 is blurred and we do not distinguish axes label with the corresponding curves.

+ Figure 6, the x-axis is not enough detailed; a number from 1 to 12 months could be helpful.

The colours red and brown are not distinguishable. A reference to "DO" is missing in caption.

+ Figure 7 and Figure 20 can be removed, as it is a classical view of well-known systems.

+ In Figure 8 the caption and the information on the figure are incomplete: the meaning of the curve colour is not given on the left plot (is it the months represented on the right panel ?), is the theta in the caption different with the phi in the y-axis label ?, where is represented 2014 as mentioned in the caption ?

+ In Figure 9, the white-blue colorbar does not allow distinguishing current velocity classes. Please consider changing the colormap.

+ Figure 12 is blurred and impossible to read. Then, we can't connect numbers in the caption with the diagram.

+ The Figure 13 can be removed, as it does not give major information for the paper purpose.

+ On Figure 14, which year is represented and what is the depth of measurements ? The meaning of CPUE could be explicitly called back in the caption.

+ Figure 15. (a) and (b) must be added on the photo or (left) and (right) has to be added in the caption.

+ Figure 17. "chlorophyll concentration" => "chlorophyll-a concentration"

+ Figure 25 is blurred.

---

## Referee Comment (RC2) · Anonymous Referee #2 · 28 Sep 2016

*** General comments

This paper comes as an introduction about the COSYNA observing system deployed in the Northern and Adriatic Seas by a large consortium, prior to more focused scientific papers in the special issue. COSYNA is presented as an integrated and complete flexible observation system, including remote observation (satellite, radar) and in situ observations, as well as modelling tools and data assimilation techniques.

First, a long description of the areas (Northern Sea and Artic Sea) and of their circulation and hydrological patterns is given. Then, the objectives and the international context are explained, showing the diversity of potential data users (from the scientific community to operational users) and the links with various initiatives at European and international levels. Third, the different components of the system are described

in a very detailed way (stations at fixed locations, mobile platforms such as gliders or ferryboxes, satellite products, HF radar data, GPS bird tracking system, models and assimilation tools, oceanographic cruises...). In this section, the authors refer to interesting scientific results of previous papers or papers of this special issue. In the following section, a description of the development of new sensors (Alkalinity sensor, nutrient sensor, molecular observatory,...) performed in the framework of COSYNA is given. Then, data management and data products are described, as well as outreach activities and stakeholder interaction, insisting on the public and free access to data collected by COSYNA. The last section deals with the future of COSYNA, in particular its spreading toward new areas, new partners and new scientific products and research associated subjects. Overall, this paper gives a lot of details on the system and on the observed areas. The spatial and temporal coverage, the technical developments, as well as the diversity of the systems that are used, make COSYNA an impressive observing system that a lot of scientists would love to have in their research geographical area.

However, the paper claims to be exhaustive, which sometimes results in long descriptions that make parts of the paper cumbersome to read. My main concern is thus on the form of the document that requires revision. I would recommend to shorten some sections and remove some figures. Reference to other papers of the special issue should also be emphasized. A few suggestions are provided in the following comments below which may help to address this issue. Reference to other papers of the special issue could also be emphasized.
* * *
*** Specific comments:

* Section 1 :

The lists of COSYNA's partners sprays over 12 lines, which interrupts the reading. Could this information be shortened and details put in another section or in the acknowledgments? More could be said about the originality of this system compared to other existing observation systems, and about the research questions underpinning the system. This last point is only approached in Section 3.

* Section 2:

This section is dedicated to the presentation of the area of observation. A general map containing the two areas, both the North Sea and the Arctic coast, is required. It would also be good to have an idea of the bathymetry in the different areas. Figure 3 is a zoom on a particular area, it would be better to have the location of the station on a larger map. On Section 2.2 the reader has to wait until l.24 of p.7 for a figure of the area, although the same area is mentioned before at l.9. Also, the description of the two areas is too long. This section of the paper should be shortened. For example on p.5, at l.31, are the residual currents useful to the purpose of the paper? (and isn't there any tidal current residual?)

p. 5 l.14 Currents are not directly dominated by a tide (replace "M2 lunar tide" by "M2 lunar tidal component").

* Section 5 "Observations" :

The idea of this paper is to link previous works with the results presented in this special issue However, more could be made in order to emphasize the new results of the special issue.

The location of the stations are often difficult to assess (for example at l.22 p.10 or l.18 p.10) or repeated ships/gliders routes, as Figures 1 and 2 are not sufficient to locate them. Please add a figure with all the fixed platforms of table 2, and refer the reader to the figure in the text.

P.10 l.24: explain the link between tidal dynamics and matter budgets.

p.11 l. 8-9: Please clarify, as one could understand that it is the viewing angle of one radar that enables getting the surface current vectors from that sentence.

[Figure]

p.12 l.5: add "or trends" after "long term records"

p.14 l. 16: please give examples of research questions

p.15 l. 9-10: The oceanographic sensors described in Section 5.4 are O2, pH, pCO2,....are they really standard sensors?

p.17 l.15: "subsurface variables"

p.17 l.22-24: the authors list the measured variables, however among the list some are not directly measured by derived from the measure (it is the case for salinity and especially for Chlorophyll-a with the measure of fluorescence).

p.17 l.27-30: the ferrybox is a very nice system, but the maintenance constrains could be mentioned. Subsection 5.6.3: is there any result obtained yet with the FLUXSO lander?

p.25 l.4: what about glider surveys?

p.25 l.6: if the surveys observations are also used for model and remote sensing systems calibration this could be added.

* Section 6:

p.26 l.26: the reviewer does not agree that the pH is a proxy for phytoplankton and primary production, it has a strong impact on them but it is not directly linked to that quantities.

Subsection 6.7 (p.30): with this passive sampling method, how do you get rid of the influence of the vessel on the measure of metals concentration?

p.35 l.13: "many modelling studies" : please add references.

Figures 7 and 20 (glider and Scanfish pictures) do not have any additional value, I would therefore suggest to remove them. I also suggest to remove Figure 13 (may this figure or the information it contains be found in another paper?)
* * *
\*\*\* Technical comments:

The figures quality should be improved.

Replace "publically" by "publicly" wherever you mention the availability of the data (p.2 l.9, p.3 l.20, p.36 l. 13 and 17, p.37 l. 24)

There is an abusive reference to Chlorophyll-a when only fluorescence is measured, please modify.

p.12, Title of subsection 5.1: I suggest to replace "fixed-point" by "fixed station"

p.10, l. 24: "dynamics is"

p. 10 l. 30 : remove comma after "located"

p.11 l. 7: "HF radar arrays are"

p. 13 l. 10 : "and operated for more than a year"

p.13 l. 21 : "at frequency M4"

p14 l.23: "analysis of"

p.18 l.25 : typo, replace "und" by "and"

p.20 l.6-7: "both CTD and ADCP sensors, and with"

p.22 l. 14-15: "The aim was to"

p.22 l.16: remove comma after "and"

p.23 l.1: "The goal is"

p.24 l. 4: "adaptation"

p.25 l.32: "gliders were"

p.27 l. 9: "was achieved"

p.28 l.4: "analyzed" or "analysed"

p.25 l.16: "depends on factors such as"

p.35 l.2: "accounting for"

p.39 l.32: "partners"

Figure 4: please describe what the underwater unit is.

Figure 5: the labels cannot be read, this figure should be improved.

Figure 8: stratification in the y label should be the same as in the legend ("$\theta$"). The time axis on Figure 8.a mentions the month, however it is said in the figure caption that it is for years 2012 and 2014, please explain. Also, what is the legend at the bottom left of the figure about? What is "b"?

Figure 12: Improve image resolution.

Figure 14, lower panel: the colorbar labels range from 32 to 36, which is very dubious for the temperature. . .please check if this is not salinity instead. . .moreover, what is the purpose of this lower panel figure? The text does not mention it so it could be removed.

---

## Author Comment (AC1) · 29 Nov 2016

Interactive comment on "The Coastal Observing System for Northern and Arctic Seas (COSYNA)" by B. Baschek et al.

Anonymous Referee #1

AUTHORS: Many thanks for your time and very valuable comments. We have addressed them all in the revised version as explained in detail below.

REVIEWER: *** General comments The manuscript proposed by B. Baschek et al. aims giving a detailed overview of the COSYNA, an integrated pre-operational observing system in Northern and Arctic Seas. As we understand, the article introduces the Special Issue dedicated to COSYNA in Ocean Science. Following this aim, no specific scientific question is addressed in the manuscript but it is more dedicated to the description of the COSYNA components. As a main general comment, the description is too long and confusing to highlight and to describe the successful integration operated in COSYNA. The balance between the amount of details and sometimes missing information or imprecisions is harming the main idea. A suggestion could be to shorten the manuscript to emphasize more the strength and coherence of the integrated system.

AUTHORS: The document has been shortened, in particular in the introduction and description of the focus regions. Several figures have been removed.

REVIEWER: The description remains also difficult to follow for non-expert readers from the geographical region. A lot of places are mentioned without an illustration on a map. The manuscript will strongly benefit from a general map with zooms and mentioned names in the text (for example: Otzumer Balje, Jade Bay, island of Helgoland, North Frisian, Lena delta, Weser, Ems).

AUTHORS: Figure 1 has been completely redone.

REVIEWER: A final general remark is more on the "network" strategy behind COSYNA. The scientific aims and the justifications of geographical extent do not appear clearly from the manuscript.

AUTHORS: We have added text on page 3 of the manuscript to clarify the networking strategy behind COSYNA as well as to justify the choice of its geographical extent.

REVIEWER: *** Specific comments * Abstract p. 2 / l. 2-3 - Authors mention that COSYNA is designed also to "assess the impact of anthropogenically induced change". This assessment is not directly provided by the network but after complex scientific analysis of the collected data. The sentence should be modified.

AUTHORS: Thank you. The sentence has been modified The automated observing and modelling system COSYNA is designed to monitor real time conditions, provide short-term forecasts, data and data products to help assess the impact of anthro-

pogenically induced change.

REVIEWER: * 1. Introduction p. 3 / l. 28-32 and p.4 / l. 1-7 - The list of contributors could be presented as a table to improve the sentence.

AUTHORS: Now moved to table 1.

REVIEWER: p. 4 / l. 9 - The paper does not introduce scientific studies but just illustrate the observation collected.AUTHORS: That is correct. It is an overview article introducing the entire Observing system. This has been clarified in the text.

REVIEWER: p. 4 / l. 10 - The "volume", I guess the Special Issue is just mentioned here. It should be mentioned before and more explicitly.

AUTHORS: A sentence has been added: The present Ocean Science and Biogeochemistry inter-journal special issue "COSYNA: integrating observations and modeling to understand coastal systems" collects contributions that highlight various aspects of the complex observing system.

REVIEWER: p. 4 / l. 11-15 - The structure of the paper needs to be given more explicitly. At least, different main parts must be linked to the section numbers.

AUTHORS: Section numbers are now explicitly given.

REVIEWER: * 2. Coastal focus regions General comment: This section should be strongly reduced. A map could give an overview of the two regions. Some details in this context are not useful as, for example: - UNESCO reference - p. 5 / l.4-5 - p. 5 / l.21-26 - p. 6 / l.4-8 - p. 7 / l.4-8

AUTHORS: We have shortened the whole chapter. The residual coastal current paragraph is not included in the new version. The overview map has been completely redone. The UNESCO reference is now moved to the paragraph describing conflicting uses

REVIEWER: p. 5 / l. 6 - The North Sea should be describe before the German Bight.

AUTHORS: The order has been changed.

REVIEWER: * 3. Objectives and Benefits p. 8 / l. 14-15 - I don't see how numerical model can integrate observations from turbulence (and from minutes) as it is scales which are not taken into account in assimilation (observations are generally degraded to be assimilated). This sentence must be clarified.

AUTHORS: This has been clarified: Numerical models of various resolutions are used to provide context for observations ranging from the turbulent to basin wide spatial scales while bridging time periods from minutes to decades. Observations are integrated into models using data assimilation techniques for resolutions, time-scales and quantities where such integration is possible and useful.

REVIEWER: * 4. International context p. 9 / l. 20-27 - These statements are not necessary for the manuscript.

AUTHORS: This paragraph has been removed.

REVIEWER: * 5. Observations p. 10 / l. 23 - What is the meaning of "(3)" and "(1)" ? Same for "(1)" and "(2)" in p. 11 / l. 8

AUTHORS: This has been clarified in text.

REVIEWER: p. 11 / l. 10-11 - The mention to CDOM, yellow substances, "gelbstoff" should homogenize in the manuscript. I would suggest keeping Coloured Dissolved Organic Matter (CDOM) as it is a name more commonly used.

AUTHORS: Gelbstoff, or yellow substance, comprise the dissolved absorbers (CDOM) as main part and the particulate. From remote sensing perspective these components are hardly distinguishable, but we always use the appropriate or both terms. At the first occurrence of CDOM on page 24 this is mentioned now.

REVIEWER: p. 11 / l. 30 - For the quality control processes, do you have references or standards to refer ?

AUTHORS: This information has been added: Quality control processes are applied and data are flagged accordingly following SeaDataNet definitions http://seadatanet.maris2.nl/v_bodc_vocab/browse.asp?order=entrykey&l=L201

REVIEWER: p. 16 / l. 29-30 - Could the authors explain more clearly the "order of 10m" ?

AUTHORS: This has been changed to: The radar signal propagates along the ocean surface beyond the horizon and is backscattered by surface waves with wave lengths between 5 and 50 m (half the electromagnetic wave length of the radar).

REVIEWER: p. 17 / l. 8 - The notion of "fusion" is not straightforward or I do not understand what is meant here. Authors should detail a bit more here.

AUTHORS: This has been clarified: Since 2013, the HF radar network is also used for ship detection, tracking, and fusing information of the radars with other sources of ship information such as from the Automated Identification System.

REVIEWER: p. 18 / l. 2-3 - The statement is already said before in the manuscript.

AUTHORS: Thank you. Sentence has been removed.

REVIEWER: 5.5 Underwater-Node System - It is not clear to me. What is the depth of the system ?

AUTHORS: The depth is now stated.

REVIEWER: p. 21 / l. 13 - Typical deployments times exceed 25 h but until how much time, it can be extended ?

AUTHORS:The sentence has been extended with: "Typical deployment times exceed 25 h to account for the diurnal inequality in tidal variations. This can be extended to longer periods (weeks) depending on measing frequency, battery and storage limitations, and the increasing risk of damage by trawlers."

REVIEWER: * 6. Sensor and Instrument Development p. 26 / l. 26 - pH is not only a proxy for phytoplankton and primary production as the water pH is not only driven (even if largely influenced) by biology.

AUTHORS: This statement has been changed to "pH can be used to estimate a system's state in terms of phytoplankton and primary production in regions of high biological activity, one of four parameters characterizing the oceanic inorganic carbon system, and an indicator for the increasing acidification of sea water."

REVIEWER: * 7. Modelling and Data Assimilation Are FerryBox data assimilated in COSYNA modelling system ?

AUTHORS: The data are used for model validation (Petersen et al., 2011; Haller et al., 2015) and assimilation studies (Stanev et al., 2011; Grayek et al., 2011; Fig. 11). This short information about FerryBox data asssimilation is provided in FerryBox part of the paper.

REVIEWER: What are the forecast periods for hydrodynamics ? It is detailed for waves but not for transport.

AUTHORS: The hydrodynmical forcecast period is 12 h as now stated in the document. FerryBox data are not pre-operationally assimilated in COSYNA modelling system. This issue is described in 1. Stanev E. V., Schulz-Stellenfleth, J., Staneva, J., Grayek, S., Grashorn, S., Behrens, A., Koch, W., and Pein, J.: Ocean forecasting for the German Bight: from regional to coastal scales, Ocean Sci., 12, 1105-1136, doi:10.5194/os-12-1105-2016, 2016. 2. Grayek, S., Staneva, J., Schulz-Stellenfleth, J., Petersen, W., and Stanev, E. V.: Use of FerryBox surface temperature and salinity Stanev measurements to improve model based state estimates for the German Bight, J. Marine Syst., 88, 45–59, 2011.

REVIEWER: p. 35 / l. 4 - I do not agree that it is "reproduced to a remarkable degree". The model is able to reproduce a deep chlorophyll maximum but its intensity and extent is not similar with observations.

AUTHORS: This has been changed as follows: Using an ecosystem model that includes turbidity fields, estimated from Scanfish observations (Section 5.9), and accounts for the acclimation capacity of phytoplankton, spatial variability in chlorophyll-a can be reproduced to a high degree (Fig. 26; Wirtz and Kerimoglu, submitted). Previous modeling attempts such as of van Leeuwen et al (2013) or Schrum et al (2006) do not capture the extreme vertical squeezing of chlorophyll-a within thin layers. Our new model results also reveal how reconstructed pelagic patterns decouple from benthic respiration patterns. Vertical deposition of freshly produced material greatly varies within the coastal ocean. In a few, mostly deeper regions, deposition prevails over resuspension, leading to depositional hotspots (Wirtz et al, in prep).

REVIEWER: *** Minor and technical corrections * 2. Coastal focus regions p. 4 / l.25 - 26 - Sentences could be rephrased ... "It is .... It is ....".

AUTHOR: The suggested revision has been made

REVIEWER: p. 6 / l.24-25 - "a-1" needs to be placed by "y-1".

AUTHOR: The suggested revision has been made

REVIEWER: * 3. Objectives and Benefits p. 8 / l. 30 - "radar" has to replaced by "HF radar".

AUTHOR: The suggested revision has been made

REVIEWER:* 4. International Context p. 9 / l. 31-32 - "to providing ... and carrying" to be replaced by "to provide ... and carry"

AUTHOR: The suggested revision has been made

REVIEWER:* 5. Observations p. 11 / l. 30 - CODM website could be given.

AUTHOR: The suggested revision has been made p. 12 / l. 2 - The number (4 or 5) of fixed stations can be explicitly written. AUTHORS: now provided (6)

REVIEWER: p. 12 / l. 22-23 - Please homogenize the writing. In this part we find back past verb when present is used in other parts.

AUTHORS: That is unfortunately hard to avoid since some units are not operational while others are still in use. Please refer to Table 3 and 4 for details. The writing has been homogenized were possible.

REVIEWER: p. 13 / l. 4-6 - The maintenance frequency could be given.

AUTHORS: We would like to keep the statement as is. We adapted the aimed intervals to the biological productivity, i.e. four times per month from mid of May to mid of September and two times per month during spring and autumn. Due to the dependency on the wave conditions, the planned maintenance frequency could generally not achieved. To explain this would make the text quite lengthy. Our intention here was to give an impression about the effort that is needed to operate a pole in this area.

REVIEWER: p. 16 / l. 14 - "The measurements taken with COSYNA gliders is ...." => " ... gliders are ...". p. 16 / l. 17 - "The data was ..." => "The data were ...". p. 17 / l. 1 - "The Systems ..." => "The systems ...". p. 18 / l. 25 - "und ..." => "and ...". p. 21 / l. 15 - The acrnomy ADCP should be mentioned in brackets before to be used. p. 21 / l. 21 + p. 22 / l. 14 - The way of writing in situ (in situ, in-situ) must be homogeneous in the manuscript. p. 23 / l. 21 - "chlorophyll"-a concentration. p. 24 / l. 29 - "chlorophyll-a" => "chlorophyll-a concentration". p. 25 / l. 28 - PSICAM is not defined at this stage in the manuscript. * 6. Sensor and Instrument Development p. 28 / l. 15-16 - "phytoplankton fluorescence" => "fluorescence". p. 28 / l. 16 - "dependents" => "depends". p. 30 / l. 4 - "Fig. 22" => "Fig. 22e". REVIEWER: * Tables Table 1: REVIEWER: + "current vector" => "current" or "current velocity" AUTHORS: done REVIEWER: + "oxygen" => "dissolved oxygen" AUTHORS: done

AUTHORS: All of these minor corrections have been made

REVIEWER: Table 2: + the level/depth of measurement is missing.

AUTHORS: Since the platforms often have several sensors at multiple depths it would be hard to integrate this in an overview table. The details are provided, however, in the individual sections.

REVIEWER: + For FINO-3, is it 2016 of now ?

AUTHORS: yes, but observations have stopped in 2016.

REVIEWER:* Figures General comment: There is a gap between this manuscript and figures ready for publication because they are often blurred and some are even not readable.

AUTHORS: Agreed. Several figure have been improved.

REVIEWER: + Figure 1 is blurred in the pdf.

AUTHORS: Figure has been completely redone

REVIEWER:+ Figure 4, reference (left/right) to the pictures is missing in the caption.

AUTHORS: done

REVIEWER:+ Figure 5 is blurred and we do not distinguish axes label with the corresponding curves.

AUTHORS: The Figure has been improved

REVIEWER:+ Figure 6, the x-axis is not enough detailed; a number from 1 to 12 months could be helpful.:The colours red and brown are not distinguishable. A reference to "DO" is missing in caption. This has been changed

AUTHORS: Thank you. This has been changed

REVIEWER: + Figure 7 and Figure 20 can be removed, as it is a classical view of well-known systems.

AUTHORS: Both Figure have been removed

REVIEWER: + In Figure 8 the caption and the information on the figure are incomplete: the meaning of the curve colour is not given on the left plot (is it the months represented on the right panel ?), is the theta in the caption different with the phi in the y-axis label ?, where is represented 2014 as mentioned in the caption ?

AUTHORS: The figure caption has been clarified.

REVIEWER:+ In Figure 9, the white-blue colorbar does not allow distinguishing current velocity classes. Please consider changing the colormap.

AUTHORS: The figure has not been changed yet, but will be redone before final submission.

REVIEWER: + Figure 12 is blurred and impossible to read. Then, we can't connect numbers in the caption with the diagram.

AUTHORS: The figure has been completely redone.

REVIEWER: + The Figure 13 can be removed, as it does not give major information for the paper purpose.

AUTHORS: Figure has been removed

REVIEWER:+ On Figure 14, which year is represented and what is the depth of measurements ? The meaning of CPUE could be explicitly called back in the caption.

AUTHORS: Caption has been redone: "Upper panel: The temporal abundances of the main biota groups assessed with a stereo-optic sensor attached to the Underwater-Node System in Spitsbergen from January 2014 to March 2014. CPUE (catch per unit effort) refer to total number of organisms per group counted per week. Lower panel: The temporal and spatial pattern of salinity in the depth range between 0 to 10 m assessed with one remote controlled vertical CTD profile per day during the same time period when the biota measurements (upper panel) were done."

REVIEWER: + Figure 15. (a) and (b) must be added on the photo or (left) and (right) has to be added in the caption.

AUTHORS: done

REVIEWER:+ Figure 17. "chlorophyll concentration" => "chlorophyll-a concentration"
AUTHORS: changed

REVIEWER: + Figure 25 is blurred.

AUTHORS: The figure has been redone.

Please also note the supplement to this comment:
http://www.ocean-sci-discuss.net/os-2016-31/os-2016-31-AC1-supplement.pdf

**Supplement:**

[revised manuscript text omitted]

---

## Author Comment (AC2) · 29 Nov 2016

Interactive comment on "The Coastal Observing System for Northern and Arctic Seas (COSYNA)" by B. Baschek et al.

Anonymous Referee #2

AUTHORS: Many thanks for your time and very valuable comments. We have addressed them all in the revised version as explained in detail below.

REVIEWER: *** General comments This paper comes as an introduction about the COSYNA observing system deployed in the Northern and Adriatic Seas by a large consortium, prior to more focused scientific papers in the special issue. COSYNA is presented as an integrated and complete flexible observation system, including remote

[Figure]

observation (satellite, radar) and in situ observations, as well as modelling tools and data assimilation techniques. First, a long description of the areas (Northern Sea and Artic Sea) and of their circulation and hydrological patterns is given. Then, the objectives and the international context are explained, showing the diversity of potential data users (from the scientific community to operational users) and the links with various initiatives at European and international levels. Third, the different components of the system are described in a very detailed way (stations at fixed locations, mobile platforms such as gliders or ferryboxes, satellite products, HF radar data, GPS bird tracking system, models and assimilation tools, oceanographic cruises: : :). In this section, the authors refer to interesting scientific results of previous papers or papers of this special issue. In the following section, a description of the development of new sensors (Alkalinity sensor, nutrient sensor, molecular observatory,: : :) performed in the framework of COSYNA is given. Then, data management and data products are described, as well as outreach activities and stakeholder interaction, insisting on the public and free access to data collected by COSYNA. The last section deals with the future of COSYNA, in particular its spreading toward new areas, new partners and new scientific products and research associated subjects. Overall, this paper gives a lot of details on the system and on the observed areas. The spatial and temporal coverage, the technical developments, as well as the diversity of the systems that are used, make COSYNA an impressive observing system that a lot of scientists would love to have in their research geographical area. However, the paper claims to be exhaustive, which sometimes results in long descriptions that make parts of the paper cumbersome to read. My main concern is thus on the form of the document that requires revision. I would recommend to shorten some sections and remove some figures. Reference to other papers of the special issue should also be emphasized. A few suggestions are provided in the following comments below which may help to address this issue. Reference to other papers of the special issue could also be emphasized.

AUTHORS: The document has been shortened, in particular in the introduction and description of the focus regions. Several figures have been removed.

****************** REVIEWER: *** Specific comments: * Section 1 : The lists of COSYNA's partners sprays over 12 lines, which interrupts the reading. Could this information be shortened and details put in another section or in the ac- knowledg-ments?

AUTHORS: Information has been moved to Table 1

REVIEWER: More could be said about the originality of this system compared to other existing observation systems, and about the research questions underpinning the sys-tem. This last point is only approached in Section 3.

AUTHORS: We have tried to highlight this more at the beginning of the paper * Section 2: This section is dedicated to the presentation of the area of observation. A general map containing the two areas, both the North Sea and the Arctic coast, is required. It would also be good to have an idea of the bathymetry in the different areas. Figure 3 is a zoom on a particular area, it would be better to have the location of the station on a larger map.

AUTHORS: Figure 1, showing a map has been completely redone.

REVIEWER: On Section 2.2 the reader has to wait until l.24 of p.7 for a figure of the area, although the same area is mentioned before at l.9. Also, the description of the two areas is too long. This section of the paper should be shortened. For example on p.5, at l.31, are the residual currents useful to the purpose of the paper? (and isn't there any tidal current residual?)

AUTHORS: We have shortened the whole chapter. The residual coastal current para-graph is not included in the new version. The figure is mentioned earlier.

REVIEWER:p. 5 l.14 Currents are not directly dominated by a tide (replace "M2 lunar tide" by "M2 lunar tidal component").

AUTHORS: done

REVIEWER: * Section 5 "Observations" : The idea of this paper is to link previous works with the results presented in this special issue However, more could be made in order to emphasize the new results of the special issue.

AUTHORS: Since this paper is meant to be an overview paper that comprises the various aspects of the observing system ranging from operational observations, to sensor development and data management, outreach, etc. it would be a misbalance to emphasize the new (scientific) results more than addressed in the single sections and associated publications. The location of the stations are often difficult to assess (for example at l.22 p.10 or l.18 p.10) or repeated ships/gliders routes, as Figures 1 and 2 are not sufficient to locate them. Please add a figure with all the fixed platforms of table 2, and refer the reader to the figure in the text.

AUTHORS: Figure 1, showing a map has been completely redone.

REVIEWER: P.10 l.24: explain the link between tidal dynamics and matter budgets.

AUTHORS: This has been clarified: Starting from the Wadden Sea coast line, four stationary systems were installed on poles placed in three tidal basins of the East Frisian and one in the North Frisian Wadden Sea. They provide highly resolved measurements of the tidal dynamics for the COSYNA standard parameters (s. Table 2) and allow the integration of energy and matter budgets over the sampled catchment areas.

REVIEWER: p.11 l. 8-9: Please clarify, as one could understand that it is the viewing angle of one radar that enables getting the surface current vectors from that sentence.

AUTHORS: This has been clarified: Two HF radar arrays are installed at the North Frisian and one at the East Frisian coast with nearly rectangular viewing angle to the other two systems.

REVIEWER: p.12 l.5: add "or trends" after "long term records"

AUTHORS: done

REVIEWER: p.14 l. 16: please give examples of research questions

AUTHORS: Several examples have been provided

REVIEWER: p.15 l. 9-10: The oceanographic sensors described in Section 5.4 are $O_2$, pH, $pCO_2$,: : :.are they really standard sensors?

AUTHORS: the word "standard" was removed.

REVIEWER: p.17 l.15: "subsurface variables"

AUTHORS: changed to near-surface variables

REVIEWER: p.17 l.22-24: the authors list the measured variables, however among the list some are not directly measured by derived from the measure (it is the case for salinity and especially for Chlorophyll-a with the measure of fluorescence).

AUTHORS: We agree and have modified the sentence accordingly: "The recorded variables include temperature, conductivity, salinity (derived from temperature and conductivity), chlorophyll-a fluorescence, turbidity, dissolved oxygen (DO), the partial pressure of $CO_2$ ($pCO_2$), pH, alkalinity, nutrients, and algal groups (derived from patterns of algal fluorescence by excitation at different wavelengths)."

REVIEWER: p.17 l.27-30: the ferrybox is a very nice system, but the maintenance constrains could be mentioned.

AUTHORS: A sentence has been added: "Due to a self-cleaning mechanism, the system maintenance intervals can be extended up to several months."

REVIEWER: Subsection 5.6.3: is there any result obtained yet with the FLUXSO lander?

AUTHORS: The Lander is a recent development. First results can be found in (Figure 16; Friedrich et al., 2016; Neumann et al 2016; Ahmerkamp under review as now specified in the paper.

REVIEWER: p.25 l.4: what about glider surveys?

AUTHORS: this has been added

REVIEWER: p.25 l.6: if the surveys observations are also used for model and remote sensing systems calibration this could be added.

AUTHORS: The in-situ values measured continuously by COSYNA are qualitative not yet usable for remote sensing validation. High precision measurements of Chl-a and TSM were performed for that purpose on selected cruises that are not part of COSYNA yet. A sentence and reference for the modelling has been added.

REVIEWER: * Section 6: p.26 l.26: the reviewer does not agree that the pH is a proxy for phytoplankton and primary production, it has a strong impact on them but it is not directly linked to that quantities.

AUTHORS: This statement has been changed to "pH can be used to estimate a system's state in terms of phytoplankton and primary production in regions of high biological activity, one of four parameters characterizing the oceanic inorganic carbon system, and an indicator for the increasing acidification of sea water."

REVIWER: Subsection 6.7 (p.30): with this passive sampling method, how do you get rid of the influence of the vessel on the measure of metals concentration?

AUTHORS: This is now described in the manuscript "Normaly, the pumped water intake systems is installed at the bow of the ship hull several meters below the sea level thus ensuring that the sampled water body is continuously exchanged due to the movement of the ship and the water is not contaminated by the metal construction of the ship. Alternatively, a metal free pump system can be deployed on a crane several meters away from the ship hull."

REVIEWER: p.35 l.13: "many modelling studies" : please add references.

AUTHORS: References have been added: To include these vertical patterns into
modeling studies requires sophisticated formulations like those by Behrenfeld and Falkowski (1997) or Behrenfeld et al. (2005).

REVIEWER: Figures 7 and 20 (glider and Scanfish pictures) do not have any additional value, I would therefore suggest to remove them.

AUTHORS: Both figure have been removed

REVIEWER: I also suggest to remove Figure 13 (may this figure or the information it contains be found in another paper?)

AUTHORS: Figure has been removed

REVIEWER:*** Technical comments: The figures quality should be improved. Replace "publically" by "publicly" wherever you mention the availability of the data (p.2 l.9, p.3 l.20, p.36 l. 13 and 17, p.37 l. 24) There is an abusive reference to Chlorophyll-a when only fluorescence is measured, please modify. p.12, Title of subsection 5.1: I suggest to replace "fixed-point" by "fixed station" p.10, l. 24: "dynamics is" p. 10 l. 30 : remove comma after "located" p.11 l. 7: "HF radar arrays are" p. 13 l. 10 : "and operated for more than a year" p.13 l. 21 : "at frequency M4" p14 l.23: "analysis of" p.18 l.25 : typo, replace "und" by "and" p.20 l.6-7: "both CTD and ADCP sensors, and with" p.22 l. 14-15: "The aim was to" p.22 l.16: remove comma after "and" p.23 l.1: "The goal is" p.24 l. 4: "adaptation" p.25 l.32: "gliders were" p.27 l. 9: "was achieved" p.28 l.4: "analyzed" or "analysed" p.25 l.16: "depends on factors such as" p.35 l.2: "accounting for" p.39 l.32: "partners" AUTHORS: All of these technical corrections have been made. Thank you!

REVIEWER: Figure 4: please describe what the underwater unit is.

AUTHORS: Description has been improved in text and is referred to in the caption.

REVIEWER: Figure 5: the labels cannot be read, this figure should be improved.

AUTHORS: The Figure has been improved

REVIEWER: Figure 8: stratification in the y label should be the same as in the legend ("_"). The time axis on Figure 8.a mentions the month, however it is said in the figure caption that it is for years 2012 and 2014, please explain. Also, what is the legend at the bottom left of the figure about? What is "b"? The figure caption has been clarified.

AUTHORS: Figure 12: Improve image resolution.

REVIEWER: Figure 14, lower panel: the colorbar labels range from 32 to 36, which is very dubious for the temperature: : :please check if this is not salinity instead: : :moreover, what is the purpose of this lower panel figure? The text does not mention it so it could be removed.

AUTHORS: Caption has been redone: Upper panel: The temporal abundances of the main biota groups assessed with a stereo-optic sensor attached to the Underwater-Node System in Spitsbergen from January 2014 to March 2014. CPUE (catch per unit effort) refer to total number of organisms per group counted per week. Lower panel: The temporal and spatial pattern of salinity in the depth range between 0 to 10 m assessed with one remote controlled vertical CTD profile per day during the same time period when the biota measurements (upper panel) were done.

Please also note the supplement to this comment:
http://www.ocean-sci-discuss.net/os-2016-31/os-2016-31-AC2-supplement.pdf

**Supplement:**

[revised manuscript text omitted]